# Emergent Communication under Varying Group Sizes and Connectivities

**Jooyeon Kim**
thingsflow
Seoul, Korea
jyscardioid@gmail.com

**Alice Oh**
KAIST
Daejeon, Korea
alice.oh@kaist.edu

## Abstract

Recent advances in deep neural networks allowed artificial agents to derive their own emergent languages that promote interaction, coordination, and collaboration within a group. Just as we humans have succeeded in creating a shared language that allows us to interact within a large group, can the emergent communication within an artificial group converge to a shared, agreed language? This research provides an analytical study of the shared emergent language within the group communication settings of different sizes and connectivities. As the group size increases up to hundreds, agents start to speak dissimilar languages, but the rate at which they successfully communicate is maintained. We observe the emergence of different dialects when we restrict group communication to have local connectivities only. Finally, we provide optimization results of group communication graphs when the number of agents one can communicate with is restricted or when we penalize communication between distant agent pairs. The optimized communication graphs show superior communication success rates compared to graphs with the same number of links, as well as the emergence of hub nodes and scale-free networks.

## 1 Introduction

Communication plays a vital role for us humans to interact with one another [Austin, 1975, Allwood, 1976, Linell, 2009]. We acquire new information about the environment and other people around us through communication, which further facilitates collaboration or competition and allows us to coordinate our future decisions and actions better.

Over the past decades, numerous research outcomes have succeeded in mimicking human behavior and making artificial agents equipped with interactive communication abilities [Batali, 1998, Cangelosi and Parisi, 2002, Christiansen and Kirby, 2003, Steels, 2003, Wagner et al., 2003]. The communication in the artificial system is emergent in a way that there are no pre-specified usage rules, semantics, or syntax, all of which are formed by the artificial agents upon their necessities [Lazaridou et al., 2017]. Not only does empowering agents with their own communicative abilities set an important milestone in interactive AI systems, it also brings us steps closer to the world where machines cooperate with humans [Mikolov et al., 2016, Crandall et al., 2018, Lazaridou and Baroni, 2020].

In human societies, hundreds of millions of people use the shared, agreed communication protocols to interact. As the group becomes more complex with increased size and connectivity, communication becomes more and more vital for us to coordinate with one another. Fortunately, we humans excel at using languages for large-group coordination and problem-solving [Tomasello, 2010, Lupyan and Bergen, 2016]. However, in the AI system counterpart, the question of how emergent communication will function under different group settings is understudied. As the group becomes more complex, will emergent communication become more effective for the system in promoting group coordination, as in human society, or will it lose its functionality? Will the emergent communication still converge

35th Conference on Neural Information Processing Systems (NeurIPS 2021).

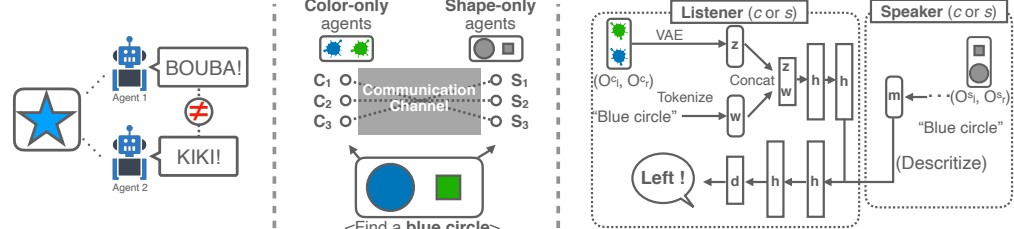

Figure 1: Left: A scenario in which two agents speak different languages to describe the same object. Middle: Bi-partite group communication using a variant of the referential game, involving color-only agents $c$ and shape-only agents $s$. Right: Communication model architecture between speaker and listener agents. Both color-only agent $c$ and shape-only agent $s$ can be speakers and listeners.

to a shared, agreed language? In the worst case, efficient interaction among agents through emergent communication can be hampered because different agents will start to use different languages to describe the same observation (the left-most panel of Figure 1).

This research provides an analytical study on emergent communication in artificial groups of different sizes and connectivities. Our goal is to investigate the effect of emergent communication and analyze how agents' general agreement on the communication protocol changes under varying group settings. To study this question, we construct settings within which communication among agents is indispensable for achieving their shared goals. Communication is their only means to interact and compensate for their limited access to information from an environment. We divide agents into two groups and allow an agent to communicate with another agent in a different group through a bi-partite communication graph.

To quantitatively evaluate the functional aspect of communication, we look at (a) the language similarity or message agreement of agents within the same group when describing the same referents and (b) the rate at which an arbitrary pair of agents in the bi-partite communication group succeed in communication and achieve the shared goal.

To this end, our contributions and findings are as follows: (1) We find that as the size of the communication graph scales up to hundreds, the language used in the group becomes less similar amongst agents. However, the communication success rate is maintained. (2) We observe that when agents only get to communicate with nearby agents, a particular form of communication peculiar to agents' locality emerges. At the same time, distant pairs of agents that have not interacted during training can still communicate when a certain threshold for the maximal communication distance is reached. (3) We present and analyze the *optimized* communication graph in terms of overall communication success rate when (a) the node outdegrees are fixed or when (b) we penalize the formation of communication links between a distant pair of agents. Compared to the fully-connected all-to-all graph, the link density decreases by $81.25\%$, but the communication success rate drops just by $5.89\%$. With random, unoptimized communication links, the success rate drops by over $40\%$ with the same link density. Throughout the optimization process, we observe the emergence of *hub* nodes taking up a large share of the total communication channels and connecting the majority of other agent nodes, as we see in the real-world scale-free social networks [Barabási and Albert, 1999].

## 2   Related work

There have been successive attempts to let agents develop their own communication protocols through emergent communication [Batali, 1998, Cangelosi and Parisi, 2002, Christiansen and Kirby, 2003, Steels, 2003, Wagner et al., 2003], followed by more recent approaches that are based on deep neural networks [Foerster et al., 2016, Sukhbaatar et al., 2016, Hausknecht, 2016, Havrylov and Titov, 2017]. [1] The idea is applied and extended to a variety of collaborative [Lowe et al., 2020, Hu and Foerster, 2020, Jaques et al., 2019, Kajić et al., 2020, Mordatch and Abbeel, 2018] and competitive [Singh et al., 2019, Jaques et al., 2019] environments. Recently, advances have been made for the emergent communication to be operationalized at scale; to prevent agents from being "lazy" and let them generate meaningful signals [Sunehag et al., 2018]; to effectively schedule

---

[1]The work by Lazaridou and Baroni [2020] surveys the nascent literature.

the communication in a limited bandwidth [Kim et al., 2019]; to foster conditions that encourage compositional and easier-to-teach languages [Li and Bowling, 2019]; to understand other agents' intention, belief, and point of view, i.e., theory of mind [Hu and Foerster, 2020]; to win a combat against another group [Singh et al., 2019]. Orthogonally, there has been attempts to make the interactive communication more dynamic and more human-like by increasing the complexity of the environment [Jaques et al., 2019, Das et al., 2019], making the communication span multiple turns [Evtimova et al., 2018, Jorge et al., 2016]. These lines of research focus on increasing the efficacy functionality of the emergent communication by suggesting more advanced perceptual, navigational, and generative AI capabilities.

On the other hand, our work falls into the category of an analytical study of understanding the emergent communication from various perspectives. Recent studies find properties of emergent language that sets itself aside from [Chaabouni et al., 2019, Kottur et al., 2017] or are shared with [Resnick et al., 2020, Baroni, 2020] natural human languages and seek for different possibilities to benefit existing NLP models [Lee et al., 2018, Lazaridou et al., 2020]. Bouchacourt and Baroni [2019] and Cao et al. [2018] studies various conditions under which the emerged language can be consonant and agreed as apposed to giving rise to idiolects or agent identity playing a crucial factor in interactive systems.

The work of Graesser et al. [2019] is especially similar to ours in that it also studies the linguistic phenomena of emergent communication at a group (community) level. Specifically, the authors formulate one, two, or five artificial communities each of which comprising agents up to size 30. By changing the inter and intra-community connection densities, the authors reproduce linguistic behaviors that can be observed in human communities. One of the findings is the emergence of the "linguistic continuum", which to a great degree resembles our finding of "dialects" in section 5.4. Our work is distinctive in a way that we do not assign agents into hard clusters of communities. Therefore, we observe finer granularities of linguistic continuums (dialects) at agent-levels rather than at community-levels. Finally, Dubova and Moskvichev [2020], Dubova et al. [2020], Tieleman et al. [2019] also studies emergent communications in group settings.

## 3 Game design

We design a communication game based on partial information. The purpose of the game is similar to that of the referential game [Lazaridou et al., 2017], which is a variant of the signaling game [Lewis, 1969]; we test whether an artificial agent can *perceive* a given observation, *speak* through the emergent protocol about the observation according to the instruction, *listen* and make the right decision. Throughout the game, the message from the speaker agent should convey crucial information about the observation for the game to succeed. Compared to the referential game, the difference is: 1) agents only get to have partial, mutually exclusive information of the observations, so the communication should solely be about describing what information each agent has from their partial, limited observation, 2) The instruction information is symmetric for both listener and speaker: both agents have access to the instruction.

In each game trial, two agents are presented with multiple pixel-based objects: a target, distractors, and instruction. The goal is to choose the correct target object for the description in the instruction. Crucially, communication is indispensable for the agents to correctly find the target since each of the two agents has its own *limitation* in the perception: One agent can only recognize different shapes, and the other can only differentiate colors. Therefore, the information of the object retrieved by the agents from their observations is mutually exclusive, but at the same time, is guaranteed to recover the complete state information when aggregated. Under this setting, the agents communicate based on their limited observations to compensate for their deficiencies and correctly choose the target object described in the instruction. In section 5.3, we derive similar findings using the more conventional referential game as well.

**Agent and action** Each of the two non-embodied agents is chosen from two distinctive *groups*. We denote agents that can only perceive colors (i.e., the *color-only* agents) by $c$, and the corresponding group by $\mathcal{G}^c$, such that $c \in \mathcal{G}^c$. Similarly, the *shape-only* agents and the corresponding group are denoted by $s$ and $\mathcal{G}^s$. The number of agents of both groups is set to be the same, i.e., $|\mathcal{G}^c| = |\mathcal{G}^s| = N$. We consider a situated environment, where the action space $\mathcal{A}$ is divided into the disjoint communication action space $\mathcal{A}^m$ and decision-making action space $\mathcal{A}^e$, such that $\mathcal{A}^m \cup \mathcal{A}^e = \mathcal{A}$ and

$\mathcal{A}^m \cap \mathcal{A}^e = \emptyset$. Of color-only and shape-only agents, one is randomly selected and act as a *speaker*; it takes its observations, i.e., objects and instruction, as inputs and sends a *message* $m \in \mathcal{A}^m$ to the other agent as an output, which can be either discrete or continuous. Next, the other agent is randomly selected and act as a *listener*. This agent takes the message along with the observations and makes a *decision* $d \in \mathcal{A}^e$. The decision is about choosing the position of the correct target object. Thus, the size of the decision making action space $|\mathcal{A}^e|$ is equal to the number of objects in the observation. Finally, the communication between agents $(c, s)$ is *successful* if the decision $d$ finds the target object. In the referential games, both speaker and listener agents get to observe the full information about the objects, i.e., both shapes and colors, but only the speaker agent has an access to the instruction and the orders of the objects are shuffled for the listener agent.

**State and observation** A state $s$ is a 3-tuple $s = (o_t, O_d, w)$ with the target object $o_t$, the distractor objects $O_d$, and the instruction $w$. Each object is of different colors and shapes [2], and the instruction tells of choosing an object with certain color and shape. Therefore, for $n$ and $m$ different colors and shapes, there can be $n \times m$ varying instructions. We represent the instruction using a fixed-sized vector by concatenating $n$ and $m$-dimensional one-hot vectors. Observations comprise objects and instructions. In section 5.3, we also conduct experiments using real-world images [Krizhevsky, 2009] instead of the artificial shape and color observations.

**Communication graph** A bipartite *communication graph* $\mathcal{G}$ is formed between $\mathcal{G}^c$ and $\mathcal{G}^s$ as multiple pairs $(c, s)$ communicate, i.e., form a set of directed *communication links* $(c, s), (s, c) \in \mathcal{E}$, and establish communication protocols.

**Message agreement and communication success rate** We introduce message agreement and communication success rates used to evaluate language convergence. First, let $a$ and $a'$ be two agents randomly sampled from the same color-only or shape-only group, i.e., $a, a' \in \mathcal{G}^c$ or $a, a' \in \mathcal{G}^s$, and let $m^a$ and $m^{a'}$ be the messages generated by the agents from the same observation $o \in \mathcal{O}$. The message agreement of a certain group, which is either color-only or shape-only, in a communication graph is measured by the expected similarity between two messages $m^a$ and $m^{a'}$: $\mathbb{E}_{a, a' \sim \mathcal{G}', o \sim \mathcal{O}}[f(m^a, m^{a'})|o]$, where $f(\cdot, \cdot)$ is a similarity measure and $\mathcal{G}' \in \{\mathcal{G}^c, \mathcal{G}^s\}$. Next, the communication success rate of a communication graph is measured as: $\mathbb{E}_{c \sim \mathcal{G}^c, s \sim \mathcal{G}^s, o \sim \mathcal{O}}[r(c, s, o)]$ with $r$ being an identity function that returns 1 if agents successfully find the target object.

## 4 Group communication model

We describe how agents communicate with others within $\mathcal{G}$ their speaking and listening networks and explain the training procedure. In a communication graph, the selected color-only agent $c \in \mathcal{G}^c$ and shape-only agent $s \in \mathcal{G}^s$ with $(c, s) \in \mathcal{E}$ or $(s, c) \in \mathcal{E}$ communicate via fixed-sized messages and make collective decisions. The network architecture is illustrated in the right panel of Figure 1. The communication algorithm that summarizes the overall procedure can be referenced in Appendix B.

**Speaker** In each trial of the game, one color-only or shape-only agent $c \in \mathcal{G}^c$ or $s \in \mathcal{G}^s$ is selected from its group and serves as a speaker, generating a message $m$. The speaker agent starts by processing a raw pixel-based observation input, consisting of multiple objects with different shapes or colors, into a fixed-sized encoding using variational autoencoder (VAE) [Kingma and Welling, 2014, Rezende et al., 2014]. [3] Each encoded vector is then concatenated with the instruction vector and is used as an input to the speaker network. The speaker network is a feedforward neural network and outputs a fixed-sized message $m$.

**Listener** The listener agent processes the observation in the same manner as the speaker network. However, rather than outputting the message, the listener network's intermediate layer is concatenated by the message generated from the speaker agent. The expanded intermediate layer is then used as an input for the upper layers of the network. It outputs the decision action $d$ of size $|\mathcal{A}^e|$, i.e., predicted position of the target object.

---

[2]We also differentiate the size, rotation, and angle of objects and vary the color codes within the same color to diversify the state-space, as further detailed in Appendix A.

[3]Each agent trains VAE separately so that no information is shared at this stage.

**Training** During training, we iteratively sample communication links between two agents $c$ and $s$, and apply backpropagation, with binary cross-entropy as the loss function over the training batch. We adopt centralized training and decentralized execution regime [Foerster et al., 2016]. Therefore, when two agents $c$ and $s$ communicate during training, the gradients flow through the communication channel from one agent's listening network to another speaking network. However, agents do not have access to the actual inner architecture, the parameter values of other agents, and no gradient flow during execution. Also, agents within $\mathcal{G}^c$ and $\mathcal{G}^s$ are completely decentralized. Message discretization is achieved by the Gumbel-softmax relaxation [Jang et al., 2017, Maddison et al., 2017]. Finally, we ensure that agents $c$ and $s$ communicate *enough* to learn the communication protocol and be fairly compared but not too much to prevent the generated messages from overfitting the training data by adopting the early stopping rule with the validation set. Refer to Appendix C for the detailed architectural settings and training results.

## 5  Experiments

We investigate how different configurations of communication graph affect language convergence in group communication. Specifically, we answer the following questions: 1. **Group size:** In all-to-all communication, i.e., when all agents communicate with each other, how does the size of the communication group $N = |\mathcal{G}^c| = |\mathcal{G}^s|$ affect the message agreement and communication success rate? 2. **Group connectivity:** In neighbor-to-neighbor communication, i.e., when agents only communicate with nearby agents, how successful can a distant pair of agents – who has not exchanged messages during training – communicate? How similar do their languages become? 3. **Optimization:** Can we structuralize the optimal communication graph in terms of communication success rate when agents can only establish a limited number of communication links?

### 5.1  Experimental setup

**Game configuration** Agents observe 10 objects with 1 target object and 9 distractors. The size of the decision-action space $|\mathcal{A}^e|$ is set to 10. For both discrete and continuous messages, we set the dimensionality of $m \in \mathcal{A}^m$ to 10. Discrete messages are binary vectors using the Gumbel-softmax relaxation. Continuous messages are real-valued vectors. [4] Objects observations are RGB pixel-based images with sizes $32 \times 320$. Each image contains 10 objects with 6 different colors (red, green, blue, cyan, magenta, yellow) and 5 different shapes (ellipse, triangle, quadrilateral, pentagon, hexagon). Instruction is an 11-dimensional vector that concatenates 6 and 5-dimensional one-hot vectors for shapes and colors. Example object observations are in Appendix A.

**Data** We generate 128,000 observations for a training set and 12,800 observations for test and validation sets. The early stopping rule is applied for the validation set, and the reported results are calculated using the test set. Finally, the model parameter settings are detailed in Appendix C.

### 5.2  Preliminary results and message categorization

In the absence of communication, the success rate of agents correctly identifying the target object is 0.236. After the communication, the success rate increases to 0.972, the maximal success rate the agents can achieve, as there are cases of having one or more distractors with the same shape and color as the target object. This result confirms that the messages transmitted from a speaker agent convey critical information the speaker agents have but the listener agents do not. Therefore, we can formally *categorize* all messages transmitted by the speaker agents based on their instructions and observations.

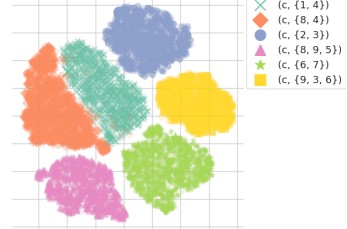

Figure 2: Message categories

**Definition 5.1.** (message categorization). Let $\{..., p_n, ...\}$ be the unique positions of the objects that have the colors (shapes) described in the instruction $w$ of a color-only agent $c$ (shape-only agent $s$), with $1 \le p_n \le |\mathcal{A}^e|$. The *category* of the message transmitted from the agent $c$ ($s$) is defined as

---

[4]We can straightforwardly restrict the range of the message values by e.g., applying the softmax or sigmoid function to the message. In our experiments, findings and statements made are consistent with and without such restrictions.

$$\left(c, \{\dots, p_n, \dots\}\right) \text{ or } \left(s, \{\dots, p_n, \dots\}\right).$$

Figure 2 validifies out message categorization. Here, we randomly sample 21,000 messages across from 32 color-only agents with 6 different categories and visualize them using $t$-SNE. Then, messages is each category are highlighted by different markers and colors. From the figure, we confirm that the messages in same categories are clustered in the latent space and exhibit high topographic similarity.

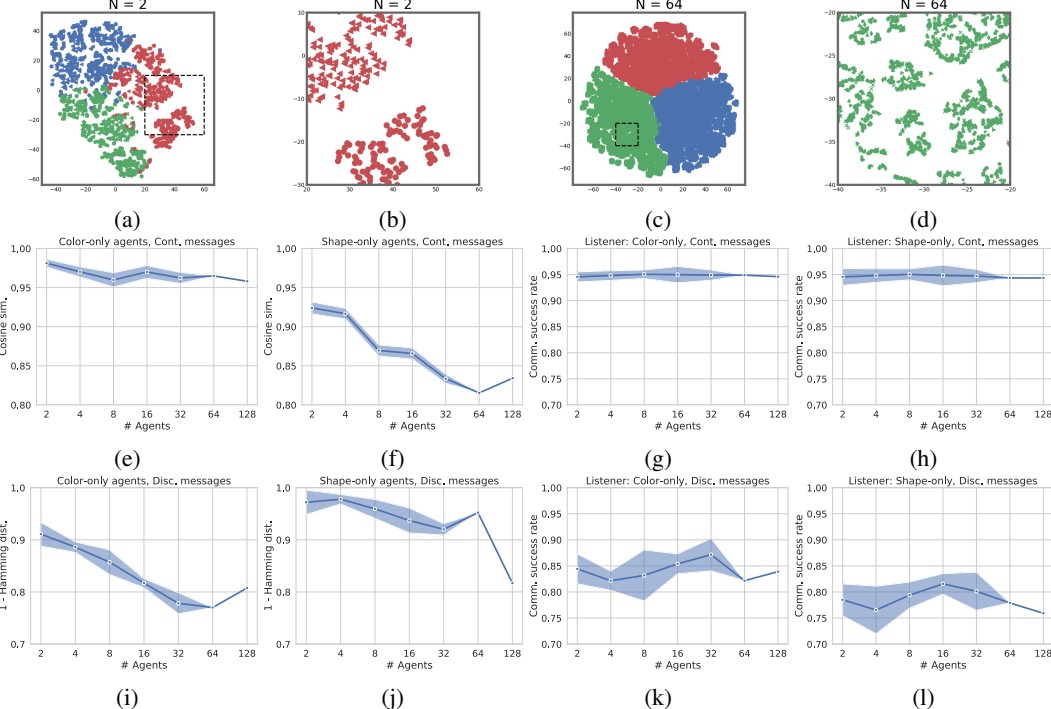

Figure 3: All-to-all communication. (a)-(d): t-SNE plots of messages. (b) and (d) are the magnified portions of (a) and (c) respectively. $N$ is the number of agents and each point is a single communication message. Colors represent different, randomly selected message categories, and markers represent different agents. As $N$ grows, *what* is in the message becomes more relevant than *who* the speaker is for identifying the message. (e)-(l): Similarities of messages sent from different agents with same observation and communication success rates of the communication graph with respect to the number of agents $N$ in the group. (e)-(h) and (i)-(l) are the results when agents use continuous and discrete messages, respectively. As $N$ grows, the messages sent by different agents become less similar, but the communication success rate is maintained.

## 5.3 Group size

We answer the first question regarding the all-to-all communication, where all agents in $\mathcal{G}^c$ communicate with all agents in $\mathcal{G}^s$. The top panels of Figure 3 illustrate the t-SNE [Maaten and Hinton, 2008] plots of the messages generated, with $N$ being the size of $\mathcal{G}^c$ and $\mathcal{G}^s$ ($|\mathcal{G}| = 2N$). We only show the color-only agents' results with a continuous message setting. [5] Each dot represents a single message, with different markers representing the messages generated by different agents. Different colors represent different, randomly selected message categories. [6] As the group size increases, the messages tend to gather up by their respective categories rather than agent identities (panels (a) and (c)), and within each cluster, the identities of the agents who transmit the messages become more difficult to be separated (panels (b) ad (d)). As $N$ increases, *what* is entailed in the message becomes more relevant when identifying the message than *who* is sending the message.

---

[5]Similar results can be obtained from the other configurations, e.g., shape-only agents with discrete messages.
[6]For the visualization purpose, we only display three categories.

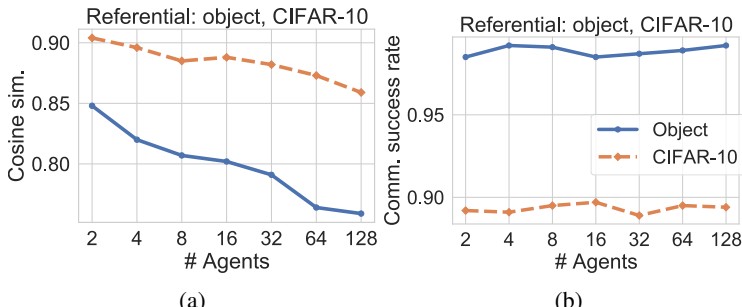

(a)                                                                                  (b)

Figure 4: All-to-all communication with the referential game setting and continuous messages. The blue solid lines are the results from the color-shape-object dataset and the orange dashed lines are the results from the CIFAR-10 dataset. (a): message similarity decreases as the number of agents within a communication graph increases. (b): communication success rate is maintained as the communication group size grows.

Second, from the subfigures (e), (f), (i), (j) of figure 3, the message agreement calculated by the cosine similarity decreases as the group size $N$ increases, for both color-only and shape-only groups and both continuous and discrete-message settings. On the other hand, in the subfigures (g), (h), (k), (l) of figure 3, the communication success rate does not increase nor decrease as $N$ increases. Even though $m^c$ and $m^{c'}$ become dissimilar as $N$ increases, the rate of the color-only agents $c$ and $c'$ successfully communicating with the shape-only agents $\forall s \in \mathcal{G}^s$ does not decrease.

In Figure 4, we observe similar trends from a different game and datasets. We analyze (a) the message similarity and (b) the communication success rate using the referential game Lazaridou et al. [2017], where agents have access to the full observation and one agent is randomly selected and act as the sender (speaker), with an access to the instruction while the counterpart receiver (listener) agent does not. Furthermore, aside from the artificial shape-color object datasets, we also experiment with the real-world CIFAR-10 images with ten classes [Krizhevsky, 2009].

Overall, as $N$ increases, messages in the same categories exhibit less topographic similarities by agents, while maintaining the communication success rate. Furthermore, agents' languages becoming more similar does not directly translate to the increased communication success rate.

## 5.4 Group connectivity

In neighbor-to-neighbor communication the color-only agents $\mathcal{G}^c = \{c_1, ..., c_N\}$ and the shape-only agents $\mathcal{G}^s = \{s_1, ..., s_N\}$ communicate with nearby-agents only, i.e., $c_i$ only communicates with $\{s_{i-j}, ..., s_{i+j}\}$ and vice versa, with $j$ being the maximum communication range. Thus, $j = 0$ signifies one-to-one communication.

Figure 5 illustrates the neighbor-to-neighbor communication results with $N$ set to 64 and varying the maximum communication range $j$. Each dot in the top panels represents a single message. Here, all messages are in a same category and different colors represent different localities of the agents; as $i$ increases in $\mathcal{G}^c = \{c_1, ..., c_i, ..., c_N\}$, the color becomes darker, from light green to black.

When $j$ is small ($j = 1$), multiple separate clusters form, and the dots of the same colors are scattered, suggesting that the agents' communication patterns are distinctive and do not depend too much on the locality. As $j$ increases ($j = 16$), the groupings of messages emerge according to agents' locality; a particular form of communication is shared by agents at the top, middle, and bottom of the graph. Simultaneously, as $j$ grows, the messages of the pairs of color-only agents that are far away from each other become more similar (the middle panels, off-diagonal parts; the row and column indices indicate the identities of color-only agents). Moreover, the communication success rate between the pair of agents that has never communicated during training increases (the bottom panels, off-diagonal parts; the row indices indicate the localities of the color-only agents and the columns are the localities of the shape-only agents).

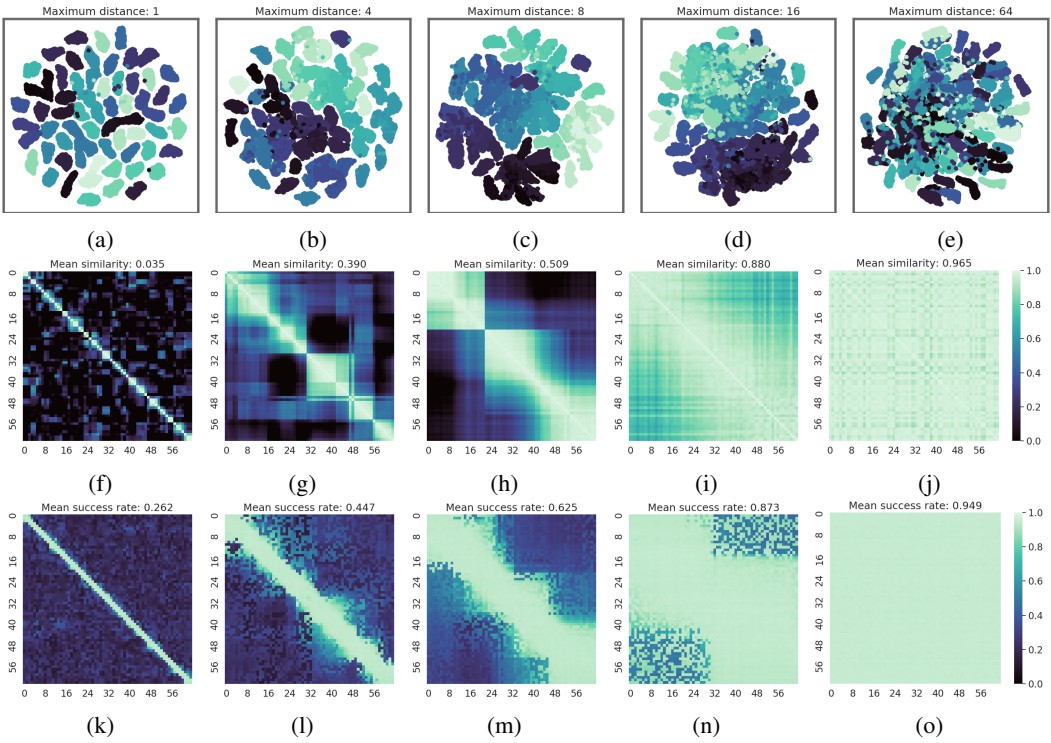

Figure 5: Neighbor-to-neighbor communication. (a)-(e): Color represents locality: lighter colors at the top of the bipartite graph and darker colors at the bottom. When the maximal communication distance ranges from 1 (a) to 16 (d), groupings of communications that are peculiar to their locality, e.g., *dialects* are formed. However, when the maximal communication distance is small (a), dialect clusters are relatively small, and such clusters from distant localities often exhibit high topographic similarity. In contrast, when the maximal communication distance is large (d), the topographic similarity of dialects is clearly reflected by the localities. At the same time, their messages become more similar ((f)-(j)) and the communication success rate increases ((k)-(o)), even between non-neighbors (off-diagonal parts).

As the maximum communication range increases, a particular form of communication peculiar to agents' locality, which is analogous to a *dialect* in human languages, emerges, and the agents' overall language gradually converges.

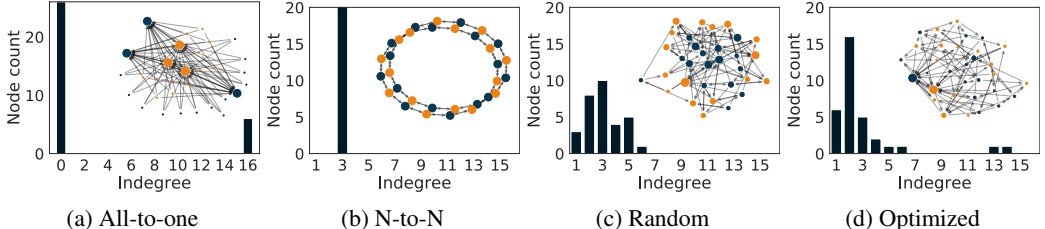

(a) All-to-one    (b) N-to-N    (c) Random    (d) Optimized

Figure 6: Different communication graphs and the respective in-degree distributions.

## 5.5 Communication graph optimization

In the previous subsection, we observe that in the sparse setting a pair of agents located far away struggles to communicate successfully and become speaking dissimilar languages. In this regard, we aim to find the *optimal* communication graph that maximizes the communication success rate. We formulate two settings: The first setting is the hard-constraint setting, with all agents share the same limited number of outgoing communication links. With $N = 16$, we restrict the number of outgoing

Table 1: Communication graph optimization. We restrict the number of outgoing edges for each agents. Reported numbers are the success rates with one standard error over five different trials.

|  | No Link | All-to-one | N-to-N | Rand. Links | Optimal |
|---|---|---|---|---|---|
| Success Rate | 0.236 | $0.384 \pm 0.031$ | $0.454 \pm 0.017$ | $0.578 \pm 0.154$ | $\mathbf{0.894 \pm 0.021}$ |

links to 3. A color (shape)-only agent can form communication links with any shape (color)-only agents but the total number of outgoing links from the agent always has to be 3. Therefore, the graph density remains constant at 0.1875 (3/16) throughout optimization.

The next setting is the soft-constraint setting, where we do not restrict the number of outgoing links but rather penalize the formation of communication links between distant agents. Specifically, we maximize the communication success rate of the communication graph subtracted by a penalty $p = k \sum_{c_i \in \mathcal{G}^c, s_j \in \mathcal{G}^s} |i - j + 1| \mathbb{1}_{(c_i, s_j) \in \mathcal{E}}$, with the parameter $k$ controlling the density of the communication links, and $|i - j + 1|$ favoring links with shorter communication distances. Therefore, there is a cost entailed when setting a communication link, and the cost increases when the distance between a pair of agents increase. We set $k = 0.0012$ so that the objective value becomes zero in the all-to-all communication, i.e., fully connected graph. For both settings, we optimize the link structure using covariance matrix adaptation evolution strategy (CMA-ES) [Hansen and Ostermeier, 2001, Hansen, 2006, 2016] with population size 32. For each link structure setting, we use the averaged score from 4 different initializations to stabilize the optimization process. Refer to Appendix D for additional details regarding the optimization process.

Table 1 shows the communication success rate of the optimized and comparison communication graphs in the hard-constraint setting. All except for no link have the same graph density of 0.1875. The visualizations of the graph structures along with their indegree histograms are shown in Figure 6. All-to-one and neighbor-to-neighbor (N-to-N) are two extreme cases. In All-to-one graph, only 3 randomly selected agents in $\mathcal{G}^c$ and $\mathcal{G}^s$ have the in-degrees of $N = 16$, while the rest of the agents have no incoming links. On the contrary, in neighbor-to-neighbor graph, all agents have the same in-degrees and out-degrees of 3. The communication success rate is less than 0.5 for both graphs. Also, when the links are formed randomly, the mean communication success rate is 0.578. Remarkably, we are able to achieve the communication success rate of 0.894 after the optimization. In all-to-all communication, the communication success rate is 0.950. The communication success rate decreases just by $5.89\%$ while the communication density is dropped by $81.25\%$. The panel (d) of Figure 6 show the visualization of the optimized communication graph; while most agents have in-degrees ranging from 1 to 4, one agent in each group $\mathcal{G}^c$ and $\mathcal{G}^s$ that has in-degree greater than 12. These agents act as the *hub* nodes, increasing the overall communication success rates. In random graphs (panel (c) of Figure 6), there is no such node with high centrality value.

Table 2: Soft-constraints: comparison with baselines.

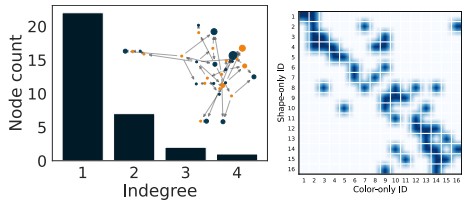

Figure 7: Optimization with soft-constraints.

|  | # Links | Penalty | Success Rate |
|---|---|---|---|
| No Link | 0 | 0 | 0.236 |
| 1-to-All | 31 | 0.094 | $0.251 \pm 0.091$ |
| N-to-N | 74 | 0.050 | $0.454 \pm 0.031$ |
| Rand. Links | 19 | 0.053 | $0.513 \pm 0.106$ |
| **Optimal** | 46 | 0.054 | $\mathbf{0.877 \pm 0.026}$ |

Table 2 shows the communication success rate of the optimized graph under the soft-constraint setting along with the comparison settings. The comparison graphs other than 1-to-All and No-link are constructed in a way that has the similar penalty value as the optimal graph. Similar to the hard-constraint setting, the success rate of the optimized graph is much greater than those of the comparison graphs with higher or similar penalty values. The left panel of Figure 7 shows the illustration of the optimized graph along with its in-degree distribution. Unlike the hard-constraint setting, we observe a scale-free graph with majority of nodes having the in-degree of one and the number of nodes drastically decreasing as the in-degree increases. The right-panel of Figure 7 shows the optimization result for the communication graph, where the blue dots in the $i^{\text{th}}$ row and $j^{\text{th}}$ column indicate the link formation between shape-only agent $s_i$ and color-only agent $c_j$. Since there is a

larger penalty for the links between distant agents, we observe that most links are formed by nearby agents. However, we sparingly observe links being formed between distant agents (off-diagonal dots in the figure), hinting that these links are crucial for maintaining the overall communication success rate despite their large costs.

## 6  Conclusion & discussion

In this work, we investigated whether the emergent language can converge in different configurations of the communication groups. We observed that when the number of group size increases, the language spoken by different agents becomes more discordant while the overall communication success rate is maintained. This is intriguing since the agents speaking similar languages does not necessarily translate to them communicating successfully.

The analyses on temporally extended and complex environment with embodied agents would be a desirable next step. Hill et al. [2020] focused on various environmental factors that lead to generalization in an embodied system. Overcoming the existing problem of multi-agent RL, such as the lazy-agent problem [Sunehag et al., 2018], the message scheduling issue [Kim et al., 2019], and joint exploration problem [Lowe et al., 2017], would also be an interesting future direction.

Understanding whether or to what extent neural agents exhibit properties that are analogous to human linguistic evolution is an interesting open question. Further, we believe that there might be some clues given from the systematicity in human linguistic evolution [Raviv et al., 2019].

Building a clear distinction between structural (grammatical) systematicity and input variability is another interesting future research topic. With this distinction, we believe that our finding (partially) aligns with what the human language studies have found. From our result with neural agents, we find that: "for larger groups, different agents start to use less similar languages to describe the same observation". This is in direct correspondence to the "linguistic variability" in human languages [Raviv et al., 2019]. In fact, human linguistic evolution studies [Gomez, 2002, Lev-Ari, 2016, Rost and McMurray, 2009, Perry et al., 2010] suggest this "input variability" as one of the factors that promote "structural systemization".

Our finding with neural agents that show a "maintained communication success rate" despite increased input variability might suggest the emergence of structural systematicity as the group size increases. However, we hesitate to make a definitive statement on this without additional experimental analyses, e.g., the actual measurement of grammatical systematicity (Raviv et al. [2019] also measure the communication success rate as one of the metrics for such systematicity).

To summarize, linguistic variability is suggested to be the key factor that promotes structural systematicity in human language studies. Our paper finds higher linguistic variability in larger, artificial neural systems and a weak sign of structural systematicity from the maintained communication success rates.

## Acknowledgments and Disclosure of Funding

We would like to thank the reviewers for taking the time and effort to review this paper and providing invaluable, constructive comments. This work was supported by the National Research Foundation of Korea(NRF) grant funded by the Korean government (MSIT) (2019R1A2C1085371).

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
