# A Dataset

We generate 128,000 images as agents' observations using python's matplotlib library Hunter [2007] for training, and 12,800 images for validation and testing. Each observation is a 32 by 320 image that displays two shapes of different colors and shapes. As shown in Figure 8, we modify the image for the color-only and shape-only agents according to the given instruction. For example, in the image, the instruction is 'Find a green circle'. In this case, we unified the shape to 'circle' for the color-only agent. Similarly, we unified the color to 'green' for the shape-only agent. Additionally, we diversity the data by endowing different sizes, orientations, locations, and hues of the objects.

For testing the generalizability beyond the training experiences, we give additional degrees of freedom to the objects, so that the objects shown are not seen during training. The detailed specifications can be reference from the code submitted in the supplementary material.

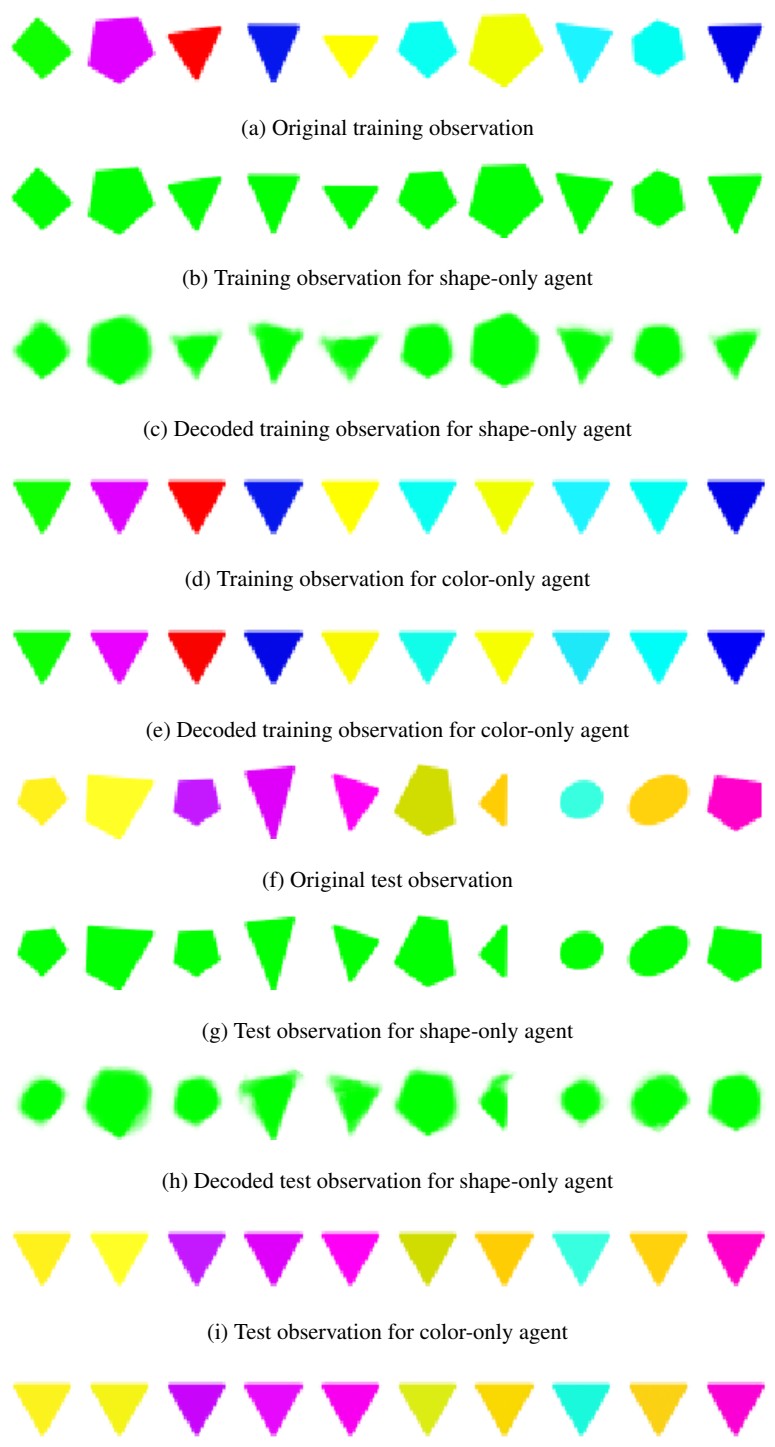

(a) Original training observation

(b) Training observation for shape-only agent

(c) Decoded training observation for shape-only agent

(d) Training observation for color-only agent

(e) Decoded training observation for color-only agent

(f) Original test observation

(g) Test observation for shape-only agent

(h) Decoded test observation for shape-only agent

(i) Test observation for color-only agent

(j) Decoded test observation for color-only agent

Figure 8: Example observations.

## B   Group communication training algorithm

Algorithm 1 details the group communication training algorithm for emergent shared multi-agent communication.

---

**Algorithm 1:** Emergent group communication

---

**Input:** Encoded vectors $z$ from the image observations $o$

1        Instruction vectors $w$

2        Batch size $B$

3        Number of agents in the group $N$

4        Training iteration $I$

5        Communication links $\mathcal{E}$

6 **for** *iter* $i = 1, \dots, I$ **do**

7     sample color-only agent $c$ and shape-only agent $s$ from $\mathcal{E}$;

8     sample the listener agent from $\{c, s\}$ ;

9     sample the training batch of size $B$ ;

10    get $B$ messages $m^c$ and $m^s$ from the speaker networks ;

11    get $B$ decision actions $d$ from the messages and the listener network ;

12    calculate the loss $\mathcal{L}$ ;

13    update the parameters of two speaker networks and one listener network ;

14 **end**

**Output:** $N$ Trained speaker and listener networks

---

## C Model architecture

Here we delineate the details on model architecture for the emergent group communication.

### C.1 Variational Autoencoder

Variational autoencoder [Kingma and Welling, 2014] is used to encode the observations. The batch size is 512, and the total number of training epochs is set to 1,000. ReLU [Nair and Hinton, 2010] and LeakyReLU (0.2) [Maas et al., 2013] are used as the activation functions for the encoder and decoder, respectively. Input is flatted 30,720-dimensional vector (32 by 320 by 3). Both encoder and decoder have one hidden layer with the dimension size being 1,024. The latent variable $z$ is a 20-dimensional vector. Finally, Adam optimizer [Kingma and Ba, 2014] is used with the learning rate being $10^{-4}$ to minimize the binary entropy error.

### C.2 Speaker and listener network

The speaker network takes the concatenation of the encoded observation image (20-dimensional) and the instruction (11-dimensional) as an input. The network has two hidden layers, each with size 256. The output (communication message) is a 10-dimensional vector. Throughout the hidden layers, ReLU is used as the activation function. In the final layer, no additional activation function is used.

The listener network takes the 10-dimensional aggregated communication messages from the color-only and shape-only agents as an input. The network has one hidden layer with size 64. The output (decision action) is a 10-dimensional vector, each feature assigned to the different positions of the predicted target object. The hidden layer uses ReLU as the activation function, and softmax is used in the final layer. For training, we set the batch size to 256. We used Adam optimizer and the binary cross entropy loss function. The learning rate is set to $10^{-4}$.

### C.3 Early-stopping

For group communication, our analytical results can be affected by the number of training epochs. For example, the reason why the message agreement is higher for $N = 8$ than for $N = 32$ might not be because of it's intrinsic group communication nature but just because the communication links for $N = 8$ have gone through more training epochs. To prevent this from happening and to prevent overfitting, we adopt the early stopping criteria for group communication settings. Specifically, early stopping is enabled when the current best accuracy on the validation set has happened before $N * patience$ epochs. Following the general practice, we early stop training by looking at the number of epochs without any progress made, where the progress is defined by the validation loss. Therefore, the patience parameter in the source code (multiagent.py) is not the actual number of training epochs but is the number of epochs after which the training should stop if the progress has not been made. The actual training epochs from Figure 9 in the appendix are actually decided by the ES criterion (based on the validation loss), not by the hard threshold. Throughout the experiments, we set the patience value to 50. Figure 9 shows the training procedures for all-to-all communication with varying $N$.

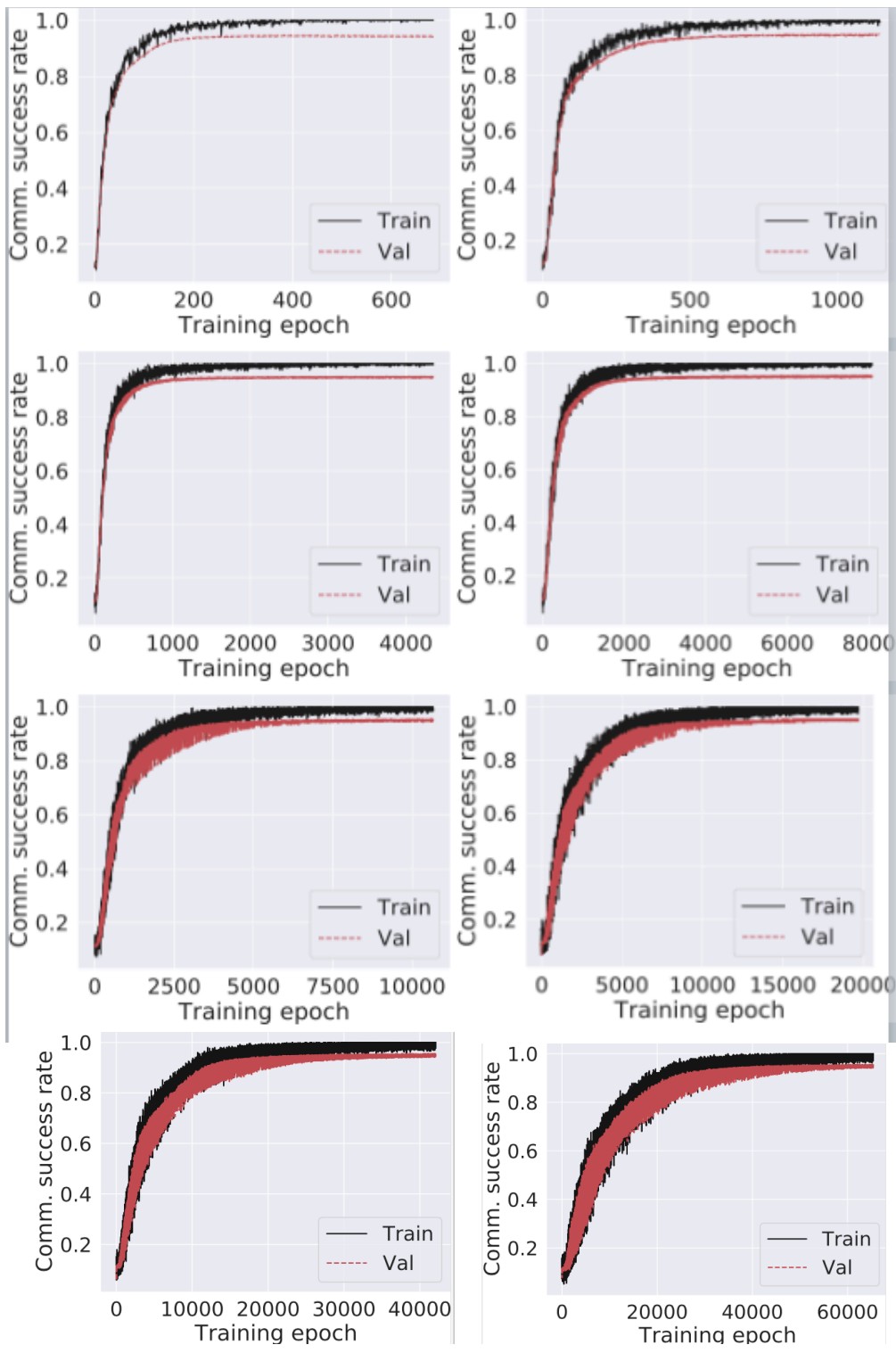

Figure 9: Training of all-to-all communication with varying $N$. Early stopping is enabled for fair comparison and to prevent over-fitting.

# D Communication link optimization

To evaluate the communication success rate for a single graph structure, in order to reduce the training time, we set the training epoch to 50. Since the communication success rate of a certain graph fluctuates according to different initializations, we used the average value over 4 different trials so as to stabilize the optimization process. Figure 10 shows the training of group communication optimization. We observe that the optimization value stabilizes at around the $1000^{\text{th}}$ iteration. With the optimized link structure, we re-trained the communication graph with higher number of training epochs and report the resulting number accordingly.

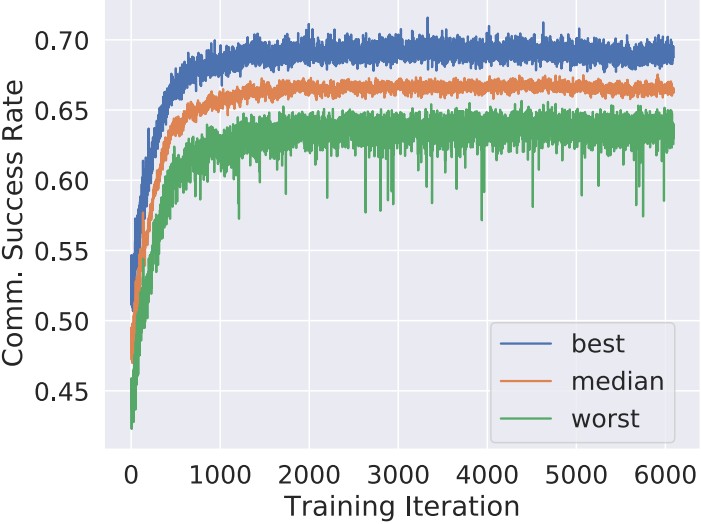

Figure 10: CMA-ES training for communication graph link structure optimization.

# E  Reproducibility - Computing infrastructure

For all group communication, we used a single GPU (GeForce RTX 2080 Ti 10GB). For CMA-ES, we used multi-CPU with the number of cores being 128, since can be straightforwardly parallelizable, and optimizing with 128 CPUs was faster than optimizing with 8 GPUs.

All codes are based on Python's PyTorch module [Paszke et al., 2019].

# F   Further discussions

In this work, we addressed the task of agents communicating in a shared, agreed language, using the emergent protocol. We acknowledge that it is extremely challenging to predict the future impact of our work in different levels and aspects, especially considering the nascent literature on emergent communication, and thereby limit our focus to:

1) The impact of emergent communication in relation to the human-in-the-loop training: In our research, our primary focus was on studying whether a universal language can be achieved in an agent system comprising of a large number of agents. Orthogonal to our research, there are some advance on emergent communication of artificial agents developing communication protocols under the human guidance [Lowe et al., 2020]. We believe that involving humans in the development of artificial agents' communication system would be an indispensable direction for understanding the agent behavior, and our endeavor of focusing on the massive setting will *conditionally* bring positive impact on promoting human-AI interactions when this direction of development is taken into consideration.

2) The impact on/from foundational research: We discussed the potential linkage between the emergent communication with a growing number of agents in the system and overparameterization in neural networks. Future development in both the emergent communication field, as well as in the foundational research regarding generalization might result in bridging the two seemingly separate branches of machine learning.