# OpenReview forum: "Emergent Communication under Varying Sizes and Connectivities"
_NeurIPS.cc/2021/Conference — NeurIPS 2021 Poster_

### Official Review · Reviewer_c94V · 2021-07-15

**Rating:** 7
**Confidence:** 4

**Summary:**

This paper proposes a number of experiments within a novel referential game where agents must learn to communicate among a population of agents. Authors design a novel referential game played in an artificial dataset of shapes and colors, where speaker and listener have access to a description of the true shape and color but are only able to either perceive shape or color. Thus, an agents' message must fill in the missing information of their corresponding listener, such that the listener can identify the correct target image.

With this setup, the authors design an experiment where there are a group of agents, and agents either are jointly trained with all other agents, or in more isolated connectivity settings (e.g. agents only speak to one or two other agents). Their aim is to see how the group size and connectivity of the group affects the learned communication protocol along a variety of axes:

- Q: how similar are the agents' utterances as the group size increases? (A: agents maintain communicative success, but messages become *more diverse* as the group size increases.)
- Q: Are agents able to generalize to new partners within a population, and to unseen tasks? (A: As agents learn from larger numbers of agents, zero-shot generalization to other agents also increases; agents do show some generalizability to unseen tasks, though it seems to vary based on group size and the complexity of the task distribution.)
- What is the optimal communicative graph which results in the least number of links between agents, but most homogenous communication?

**Limitations And Societal Impact:**

Yes

**Main Review:**

Overall, this is a fairly well-written paper with a number of interesting findings on how group size effects learned communication protocols in multi-agent environments. However, I also see several weaknesses related to the odd experimental design, imprecise evaluation, lack of (discussion of) broader applicability of these experiments and comparisons to related work. Thus, right now, I'm a bit on the fence about this paper, and am hesitant to unequivocally recommend acceptance.

## Strengths

This paper has some interesting findings:

1. Perhaps the most potentially interesting is the finding that larger community sizes increase diversity of the languages (L243-244), yet maintain linguistic success. This is perhaps the opposite of what the human linguistic evolution literature might suggest, and similar papers (e.g. Tieleman et al. 2019).
2. The experiment for identifying the optimal communication graph to maintain intelligibility among the agents is quite interesting. I imagine this might have some interesting downstream uses, but it's not quite clear to me. I'd be interested in the authors thoughts on this.

## Weaknesses

### Unusual experimental design (reference game) limits comparability to past approaches.

Authors design a variant of the reference game where agents play both the role of the speaker and listener, and are divided into agents that can only perceive *shape* and those that can only perceive *color*. In the game, unlike the typical referential game, both speaker and listener agents receive an instruction which denotes the features of the target shape, though since speaker and listener are unable to identify one of the features (shape/color) this is insufficient information to solve the task from the instruction alone. Therefore presumably the speaker must convey a message that encodes which objects have the target feature it can recognize. This is less of a referential game and more of a partial-missing-information game.

I feel like this design choice is somewhat odd and insufficiently motivated. Authors claim that this setup encourages *symmetry* of the game setting, in that agents do an equal degree of speaking and listening; but this doesn't seem unique to this experimental design, and can indeed be implemented in standard Lewis referential games (we simply endow an agent with both language understanding and generation modules). Agents with dual speaker/listener roles have also been implemented in Graesser et al., 2019, Choi et al., 2018.

In the absence of any distinct advantages of the currently presented referential game setup, I have a strong prior towards using as standard of an experimental setup as possible; this concern generally falls under the broader concern about the extent to which the findings found in this paper can be generalized to other agents, environments, and datasets.

### Imprecise evaluation

Some of the evaluation in the paper is seems vague and ad-hoc, and I would have liked more standardized/systematic ways of evaluating the emergent language. A clear example is L231, the authors group messages according to criteria identified themselves, e.g. "the target object is "blue" and it is at the 3rd position in the observation." They display only 3 types of messages, but do not state what kinds of messages were identified and vaguely state that "ther are more than three types of messages we can identify." in Footnote 6.

This is all quite confusing: it's hard to understand what criteria the authors used to categorize the messages, and how the meaning of such messages was verified (how do we know that a message means "3rd position in the observation?"). Furthermore, the authors use a t-SNE plot to illustrate that it gets harder to identify which agent transmitted what messages as the group size increases. I find it really hard to examine parts of a t-SNE plot, with 3 messages labeled according to categories that are unknown to me, and conclude based on the closeness of some of the dots in the plot that "what is entailed in the message becomes more relevant when identifying the message than who is sending the message" (and even this line is a bit inscrutable to me).

Similarly the t-SNE plots in Section 5.4, Figure 4a-e apparently plot messages for color-only speakers that "have similar observations", but again this is quite vague, and the conclusion is a bit unclear to me (how does the scattered-ness of 4e illustrate the rise of "dialects"?)

Finally, in L248, details of how the dataset tests generalization is crucial to the conclusions of this section. It's hard to reconstruct what is happening from the few sentences here ("diversify colors"). These details shouldn't be left for the Appendix.

### Artificial experiments only

Only one artificial shape dataset is explored, and the data and experimental design (as mentioned above) limit the generalizability of the present results. The dataset uses only artificial colors and shapes. Moreover, the experimental design requires cleanly dividing agents by the dataset characteristics into one of two features (color/shape)—how commonly will we encounter such a setting in practice? Do the conclusions about the regularity and systematicity of the languages based on population size extend to more tasks where it is harder to extract the relevant features (e.g. more realistic image-based reference games) or for tasks besides reference games entirely? The "simple" and "complex" tasks explored at the end of section 5.3 lack calibration - we obtain "simple tasks" by reducing the colors and shapes by one or two possibilities, but perhaps these tasks are both extremely simple in the context of other environments we might deploy our agents in. Thus, I'm hesitant to make broad conclusions that the authors do, e.g. "in complex environments, the emergent protocol generalizes better as the group size increases."

### Could use more discussion about broader impacts

There are several experimental results presented in this study, where many parameters of group communication are verified, but aside from being interesting experiments, it's somewhat unclear what the takeaways are for either people interested in linguistic evolution or people who want to build better multi-agent systems.

What are some examples we might face in practice, in multi-agent communication settings, where the communication graph of a population is not fixed, is bipartite w/ a clean division in information accessibility, etc?

The optimized communication graph experiment in section 5.5 is interesting, but it's quite unclear to me why we would want to do such a thing. For example, we find that one node emerges as "hub nodes" which regularize the communication protocol across the population. Are we meant to take away from this something about human linguistic evolution? When would we want a non-fully-connected communication graph in a multi-agent setting?

### Some overlaps and lack of comparison to related work

- Tieleman et al. Shaping representations through communication: community size effect in artificial learning systems. NeurIPS ViGIL 2019. The communication experiments with continuous messages in the authors' paper are very similar to the experiments here. Some of the conclusions of the authors should be compared/contrasted with Tieleman et al. Seems like authors conclude that languages become more discordant (but still successful) when group size increases, while Tieleman et al. almost observe the opposite: larger group sizes result in *less idosyncratic, better* learned representations.
- The Graesser et al. paper is particularly relevant; it's mentioned that there are similarities in L106, but not at all referred to for the rest of the paper, though I feel the similarities warrant more discussion. The Graesser paper is dismissed as irrelevant beacuse it "assigns agents into hard clusters of communities"  but this is a bit unclear to me - from what I recall Graesser et al. also explore communicative graphs with loosely-defined groups where there are more in-links than out-links.

## Missing references

- Tieleman et al. Shaping representations through communication: community size effect in artificial learning systems. NeurIPS ViGIL 2019
- Raviv et al. Larger communities create more systematic languages. Proc Roy Soc B, 2019. (In general I'd like some comparisons to the human linguistic evolution literature).

## Minor

- I think "varying group sizes and connectivities" would be a helpful edit to the title, as sizes/connectivities is ambiguous - I first interpreted it as size/connectivities of the architectures of speaker/listener implementations
- How exactly are the models limited to only either perceiving color and/or shape? I feel like I missed an actual discussion of this in the paper?
- L124 instruction is not a pixel-based object, I assume?
- L172 How is the VAE trained?
- L325 "...evidences the benefit of emergent communication involving larger number of participants rather than concerns." unclear sentence - what does concerns mean here? Do you mean it's better to have larger group sizes than smaller group sizes?
- [David, 1969] citation should be [Lewis, 1969]

**Time Spent Reviewing:**

3

---

> ### Author Response · Authors · 2021-08-15
> **Initial response to Reviewer c94V [1/2]**
>
> We sincerely appreciate reviewer `c94V` for giving extensive reviews. We believe that we can substantially increase the quality of our paper thanks to your comments. Below, we respond to all the individual comments/questions:
>
> ---
>
> >__Q1. Missing references and lack of comparison to related work__.
> - Thank you for pointing out the previous works that we have missed and suggesting more discussions on certain works.
>
> 1. __Graesser et al., 2019 [A]__. *"[...] I feel the similarities warrant more discussion [...]"*: We do feel that this work deserves more discussion in our paper. However, this paper is __not__ "dismissed as irrelevant" in our paper. Furthermore, our statement about the paper, "[...] assigns agents into hard clusters of communities [...]" is __very true__; although the authors do try multiple inter/intra community connectivity rates, all agents are __always assumed to be assigned into a single non-overlapping group__. In fact, their findings on the linguistic continuum are something that closely resembles our finding of dialects. However, __Figure 4 (d) of our work__ and __figure 7 of theirs__ clearly show the difference: the emergence of dialects at the level of 64 *agents* and the emergence of the "linguistic continuum" at the level of 5 *communities*. Again, we do feel that this work needs to be discussed more, especially 1) in terms of the motivational aspect: "examining the linguistic phenomena at a group (community) level" and 2) in comparison with section 5.4 (Group connectivity) of our paper. Therefore, __we will add the following paragraph/sentences in the revised manuscript__:
>   1-1. __Section 2 (Related work)__: Instead of the two sentences starting from line 105, we will create the following paragraph: "The work of Graesser et al. [2019] is especially similar to ours in that it also studies the linguistic phenomena of emergent communication at a group (community) level. Specifically, the authors formulate one, two, or five artificial communities each of which comprising agents up to size 30. By changing the inter and intra-community connection densities, the authors reproduce linguistic behaviors that can be observed in human communities. One of the findings is the emergence of the "linguistic continuum", which to a great degree resembles our finding of "dialects" in section 5.4. Our work is distinctive in a way that we do not assign agents into hard clusters of communities. Therefore, we observe finer granularities of linguistic continuums (dialects) at agent-levels rather than at community-levels."
>   1-2. __Section 5.4__: We will reiterate the similarities and differences between the two works.
>
> 2. __Tieleman et al., 2019 [B]__. *"This is perhaps the opposite of what the human linguistic evolution literature might suggest, and similar papers (e.g. Tieleman et al. 2019)", "[...] almost observe the opposite [...]"*: This is a __critical misunderstanding__. First, our finding: "messages become less similar in describing the __same observation__ as the group size increases" is __not__ the opposite of what the human linguistic evolution literature suggests: i.e., "larger communities create more systematic languages" [C] or "simpler grammar" [D]. Namely, the difference comes from a) agent-wise similarity in describing the same observation (ours) vs. b) message-wise similarity in describing different observations ([C], [D]). We thank you for suggesting the missing reference of Tieleman et al. [B] and we will cite this with added discussions, as it also focuses on the analyses of linguistic phenomena under different group (community) sizes, similar to section 5.3 of our work. However, our work on message similarity (Figures (e, f, i, j) in our paper) is __not__ the opposite of Figure 2 (left) or theirs. Again, our work is on the similarity among messages generated by different agents describing the same observation, while their work is on the similarity among messages (latent representations) describing different observations. In fact, to create a bijection between observation and message, they *average out* all messages describing the same observation, while we rather look at those individual messages and their similarities. __we will add the following paragraph/sentences in the revised manuscript__:
>   2-1: __Section 2 (Related work)__: After introducing the work of Graesser et al., 2019, we will add the following sentence: "The work of Tieleman et al., 2019 [B] also studies the linguistic behavior in different group sizes. Motivated from the findings from the human linguistic evolution literature [C, D], the authors suggest that simpler languages emerge as the population size increases (up to 16 agents). This finding relates to our finding of message similarity with respect to group size in section 5.3. The difference comes from the fact that we are interested in the degree of different agents using different messages to describe the same observation while Tieleman et al. (2019) focus on the message similarities for  different observations."
>   2-2. __Section 5.3__: We will reiterate the similarities and differences between the two works.
>
> ---
>
> >__Q2. Imprecise evaluation.__ *" [...] it's hard to understand what criteria the authors used to categorize the messages, and how the meaning of such messages was verified [...]"*:
>
> - Regarding this issue, the reviewer has pointed out three parts of the paper: 1) Line 213 of section 5.3, 2) Figure 2 (top) of section 5.3, and 3) Figure 4 (top) of section 5.4 that are hard to follow because of the informal description of message categories. First of all, we believe that this is certainly the part that needs to be improved with __more formal description__ of how we (authors) identified different message categories and explanations to validify such message categorization. With such improvement (formal categorization of messages and validifying such categorization), the three issues the reviewer mentioned can be resolved. Therefore, we respectfully believe that this comment can be better framed as "the lack of explanation on identified message categories" than "Imprecise evaluation" (we believe that there is nothing "imprecise" about our message categorization. The issue is that our explanation of such message categories is unmethodical, which made the reviewer hard to "understand"). Please let us know if our understanding of this issue is incorrect.
>
> - We will make the following revisions to resolve this issue:
>   - Formally describe our message categorization: We will replace section "5.2: Preliminary results" with "section 5.2: Preliminary results and message categorization" and add the following paragraph:
>     - In the absence of communication, the success rate of agents correctly identifying the target object is 0.236. After the communication, the success rate increases to 0.972, the maximal success rate the agents can achieve, as there are cases of having one or more distractors with the same shape and color as the target object. This result confirms that the messages transmitted from a speaker agent convey critical information the speaker agents have but the listener agents do not. Therefore, we can formally \textit{categorize} all messages transmitted by the speaker agents based on their instructions and observations.
>       - __Definition 5.1.__ (*Message categorization*). Let { $ \dots , p_n  , \dots $ }  be the unique positions of the objects that have the colors (shapes) described in the instruction $w$ of a color-only agent $c$ (shape-only agent $s$), with $1 \le n \le | \mathcal{A}^e |$. The *category* of the message $m$ transmitted from the agent $c$ ($s$) is defined as ($c, $ { $ \dots , p_n  , \dots $ }) (($s, $ { $ \dots , p_n  , \dots $ })).
>   - Validify the message categorization: Followed by this paragraph, we will validity our message categorization by showing examples of different message categories and show that messages in the same category exhibit high topographic similarity. Specifically, we will randomly sample 100,000 different messages transmitted from different agents. Then, we will randomly select 8 different message categories, e.g., ($c$, {2, 5}), ($s$, {4, 6}), ($c$, {3}) and color the messages by their categories, and __validate that the messages in the same categories are indeed clustered in the latent space__. This will be presented in the figure format with explanations (we already have the figure to be presented in the revised manuscript).
>   - Rephrase all informally explained results with the defined message categorization: We will rephrase 1) Line 213 of section 5.3, 2) Figure 2 (top) of section 5.3, and 3) Figure 4 (top) of section 5.4) with the newly defined message categorization.
>   - In response to __Q3. Unusual experimental design__, we have additional experimental results with the new game set up using the referential game. In this game, both the speaker and listener agents observe the full images (without limitations), but the instruction is given only to the speaker agent. Therefore, the critical information a message should carry is not the position of the target objects, but the actual description of the target object itself. Thus, we categorize messages by the shape and color described in the instruction. In the appendix, we will add the formal definition of message categorization with respect to this game setup.
>   - Finally, we will add explanations of how we created the extended datasets for the generalizability experiment in section 5.3 (we will NOT put this in the appendix).

---

> > ### Author Response · Authors · 2021-08-15
> > **Initial response to Reviewer c94V [2/2]**
> >
> > >__Q3. Unusual experimental design limiting comparability to past approaches.__ *"[...] encourages symmetry [...] doesn't seem unique to this experimental design", "[...] the findings [...] can be generalized [...]"*:
> >
> > - We acknowledge that our claim on the *symmetry* of the game design removing *architectural biases* is incorrect, and we will remove all our claims regarding this. For example, as pointed out by the reviewer `e6Vg`, the work of Bouchacourt and Baroni (2019) [E] designs symmetric games to reflect human-like property but this does not warrant the removal of architectural biases. We will make sure that we remove all unwarranted claims and citations.
> >
> > - We also acknowledge that we have incorrectly emphasized our novelty in the game setup. In fact, there have been numerous experimental setups that share a great degree of similarity to ours, especially on the partial observation. Our misplaced emphasis on the game setup has raised concerns on whether our findings with this "unique" setting can be applied to the existing, more "traditional" settings, and this concern is echoed by another reviewer `c94V` and `e6Vg`.
> >
> > - Our setting is not unique. In fact, partial observation with disjoint and complementary evidence given to different agents has been the key formulation that encourages the emergence of interactive communication and has been adopted by numerous previous research [F,G,H,I]. Crucially, Graesser et al. (2019) [A] studies the emergence of interactive communication in groups with the partial, disjoint, and complementary setting similar to ours.
> >
> > - The other setting that promotes emergent interactive communication between agents is the fully observed setting. This setting, referred to as the *referential game*, originates from the work of Lazaridou et al. (2017) [J] and is a variant of the *signaling game* (Lewis, 1969 [K]). Here, both agents are given the same, full observations. Communication between agents is encouraged because 1) only one agent (the speaker agent) has access to the *instruction* and 2) the *orders* or the objects displayed in the observations are different by the agents. Therefore, our setup does  __not__ correspond to the referential game, and we will make this point clear in the revised manuscript.
> >
> > - To show that our findings and claims are applicable for both settings, __we have conducted additional experiments__ with the full-observation setting (i.e., the referential game). Specifically, the color and shape variations are maintained, but all agents observe the full evidence from such objects (not color-only nor shape-only). Only the speaker agent is provided with the instruction, and the orders of the objects are shuffled in the observations of the speaker and the listener agents. In this setting, our findings are:
> >   1. The messages the speaker agents send to the listener agents changes in this setting. In the partial observation setting, the speaker agents send messages to the listener agents to inform the agent about the *positions* about the target object (Appendix D). On the other hand, in the full observation setting, since only the speaker agent has an access to the instruction and the positions of the objects are shuffled for the listener agent, the messages entail information about the *target object itself* (i.e., whether the target object is a blue circle or a red square).
> >   2. The findings and the claims made in with the partial observation setting __can also be made in the full observation setting__. Specifically, 1) in all-to-all communication, the message similarity decreases but the success rate is maintained as the group size increases and 2) the emergence of dialects in the neighbor-to-neighbor communication.
> >   - We did not conduct the experiment on the communication graph optimization with the full observation setting.
> >
> > - __Plan for the revision__: First, we will correct all claims and misplaced citations that highlight the uniqueness of our game setup. Then, in the experimental setup section, we will discuss the two commonly adopted setups for interactive emergent communication, which are: 1) partial observation and 2) full observation with instruction given to one agent only (i.e., the referential game). Then, we will state that our setting falls into the former category. Lastly, we will state in the revised manuscript that the claims/findings from the partial observation setting are echoed in the full observation setting, and __include the experimental outcomes from the full observation setting__.
> >
> > ---
> >
> > >__Q4. Artificial experiments only.__ *"Only one artificial shape dataset is explored [...]", "The dataset uses only artificial colors and shapes.", "[...] the experimental design requires cleanly dividing agents by the dataset characteristics into one of two features[...]", "[...] more realistic image-based [...]"*:
> >
> > - We have conducted an __additional experiment on a referential game with image-based dataset__. Specifically, we used the same setting as the added referential game setting (detailed in __Q3.__) but the objects are __now image-based__ (__CIFAR-10__ [L]) instead of the shape-color variations. Again, our claims regarding the population size and connectivities (sections 5.3 and 5.4) can also be made with the CIFAR-10 dataset. However, we were not able to conduct experiments on the generalizability, as we could not get an "extended dataset" as we did in the shape-color-based objects.
> >
> > - __Plan for the revision__: We will state in the revised manuscript that the claims/findings from the shape-color-based objects dataset are echoed with the image-based dataset, and __include the experimental outcomes with this dataset__.
> >
> > ---
> >
> > >__Q5. More discussion about broader impacts.__ *"[...] optimal communication graph [...] I imagine this might have some interesting downstream uses, but it's not quite clear to me. I'd be interested in the authors thoughts on this."*:
> >
> > - The emergent communication in communities has proven its efficacy in solving various tasks in interactive systems [F, G, M, N]. We agree in reviewer `8o6e` in that our optimization result would be beneficial in systems where there is a __cost entailed to form a communication link__ between agents, i.e., communication is necessary to solve a task but it is also costly. To better reflect this, we have conducted an __additional experiment__ in a new setting where we want to maximize the overall communication success rate but we also __penalize__ the addition of communication links between agents. Specifically, the penalty is based on agent localities, i.e., communication links between distant agents are penalized more than those with agents nearby. From this modified experimental setup, we have observed clearer power-law distribution along with the hub nodes in the optimized communication graph, in much resemblance to real-world human graphs (social networks). We believe that our experimental result on optimized communication graphs can motivate future research in designing communication graphs in complex, time-extended systems with embodied agents. Also, we believe that this result can provide insights into the similarity between the optimized communication graph and the real-world human social networks.
> >
> > - __Plan for the revision__: In section 5.5, we will add more discussion on the broader impact of the experiment along with the additional experimental design and results.
> >
> > ---
> >
> > >__Q6. Minor questions and comments.__
> >
> > - On the title: Thank you for the suggestion. We will change the title accordingly.
> >
> > - *"How exactly are the models limited to only either perceiving color and/or shape?"*: The limitation is directly reflected in agents' pixel-based observation, as shown in Figure 6 (appendix). We will bring some examples to the main paper (not the appendix).
> >
> > - *"instruction is not a pixel-based object?"*: Yes, it's a multi-hot vector. We will correct this in the revision.
> >
> > - L325: *"what does concerns mean here?*: Yes. We will rephrase the sentence to: "[...] since it directly evidences the benefit, rather than concerns, of involving a larger number of agents for the emergent communication.
> >
> > ---
> >
> > [A] L. Graesser et al., Emergent Linguistic Phenomena in Multi-Agent Communication Games. In EMNLP 2019
> >
> > [B] O. Tieleman et al., Shaping representations through communication: community size effect in artificial learning systems. In NeurIPS ViGIL 2019
> >
> > [C] L. Raviv et al., Larger communities create more systematic languages. Proceedings of the Royal Society B, 2019
> >
> > [D] F. Reali et al., Simpler grammar, larger vocabulary: How population size affects language. Proceedings of the Royal Society B, 2018
> >
> > [E] D. Bouchacourt, M. Baroni. Miss Tools and Mr Fruit: Emergent Communication in Agents Learning about Object Affordances. In ACL 2019
> >
> > [F] S. Sukhbaatar, A. Szlam, R. Fergus. Learning Multiagent Communication with Backpropagation. In NeurIPS 2016
> >
> > [G] J. Foerster, Y. Assael, N. de Freitas, S. Whiteson. Learning to Communicate with Deep Multi-Agent Reinforcement Learning. In NeurIPS 2016
> >
> > [H] N. Jaques, A. Lazaridou, E. Hughes, C. Gulcehre, P. A. Ortega, D. Strouse, J. Leibo, N. de Freitas. Social Influence as Intrinsic Motivation for Multi-Agent Deep Reinforcement Learning. In ICML 2018
> >
> > [I] I. Kajic, E. Aygün, D. Precup. Learning to cooperate: Emergent communication in multi-agent navigation. arXiv preprint arXiv:2004.01097, 2020.
> >
> > [J] A. Lazaridou, A. Peysakhovich, M. Baroni. Multi-Agent Cooperation and the Emergence of (Natural) Language. In ICLR 2017
> >
> > [K] D. Lewis. Convention: A philosophical study. 1969
> >
> > [L] A. Krizhevsky. Learning Multiple Layers of Features from Tiny Images. 2009.
> >
> > [M] I. Mordatch, P. Abbeel. Emergence of Grounded Compositional Language in Multi-Agent Populations. In AAAI 2018
> >
> > [N] D. Kim, S. Moon, D. Hostallero, W. Kang, T. Lee, K. Son, Y. Yi, Learning to Schedule Communication in Multi-agent Reinforcement Learning. In ICLR 2019

---

> > > ### Comment · Reviewer_c94V · 2021-08-25
> > > **Response**
> > >
> > > Thanks to authors for their detailed and high-quality response to my review, and I apologize for the slow response. The additional experiments with more standard image-based referential games are greatly appreciated, and strengthen the paper's conclusions.
> > >
> > > While I am more positive on the paper, I have a few outstanding questions:
> > >
> > > > Q1: messages become less similar in describing the same observation as the group size increases" is not the opposite of what the human linguistic evolution literature suggests: i.e., "larger communities create more systematic languages" [C] or "simpler grammar" [D].
> > >
> > > I'm still unclear as to how my claim is a crucial misunderstanding. Note that [C] reports systematicity measures with topographic similarity, which does indeed measure how similar messages are in describing different observations, *but also implicitly measures how similar messages are when describing similar/identical observations*. The idea that agents are less similar in describing *the same observation* as group size increases still seems like a *less systematic* language to me: there are fewer conventions/stability when referring to the same object, which feels less human-like (it's like a language having 10 different words that all mean "circle"). Could authors discuss why their observation that the language is less deterministic when referring to the same object (i.e. having less reliable words for categories, concepts, etc) is *not* "less systematic"?
> > >
> > > To be clear, I do *not* think that having a finding that is "opposite" existing literature is a *bad* thing, or a weakness of the paper. I admit that "opposite" was an overly strong word here: in reality, the authors' finding simply illustrates subtleties between group size and linguistic systematicity in multi-agent population, to a greater degree than is suggested by the existing literature. That is, perhaps larger group sizes result in more systematic languages, but maybe there is a breaking point where the size, or the partial connectivity of a group, leads to the formation of dialects, etc. Or indeed perhaps neural agents are not a good analog to the human language evolution literature. These would be interesting points to raise in the paper.
> > >
> > > > Q2. Imprecise evaluation
> > >
> > > I agree that a more precise and formal description would greatly improve my complaint of "imprecise evaluation." Even if authors exhaustively detail the criteria used to group messages, however, I believe the authors' proposed quantitative analyses (of verifying meaningfulness of the groups via topographic similarity) is crucial to supporting the idea that the way the authors have grouped the messages is sensible. Even if the methods are exhaustively detailed, I am naturally weary whenever authors themselves label/look at data, and so some quantitative verification of the messages clusters would be very helpful here. Could authors confirm that they have such analyses and they confirm the groupings of the authors?
> > >
> > > > Finally, we will add explanations of how we created the extended datasets for the generalizability experiment in section 5.3 (we will NOT put this in the appendix).
> > >
> > > I took a look at the appendix and I notice in L507-508 the reader is asked to refer to the code submitted in the supplementary material in order to see details of the generalizability experiments. I would really like to know the details outright (not in code) if possible—could authors provide this information in a response?
> > >
> > > This would help me evaluate the true difficulty of the generalization experiments (and what are "simple" vs "complex" tasks). As it stands I still remain skeptical of the generalizability experiments. Note this is both due to (1) lack of details provided about what we are testing and (2) the fact that generalization is only measured within a domain which I believe is overall very simple (shapes). The CIFAR experiments were a great inclusion for the rest of the paper, though it would have been great to see some sort of OOD generalization test for CIFAR too. If the paper is accepted I would suggest that authors heavily qualify their conclusion in L265-277 that "when agents are restricted with small observation and action spaces, their communication generalizes poorly as the size of the communication graph increases; in complex environments, the emergent protocol generalizes better as the group size increases." This claim is only measured within a very simple artificial shape setting and not a more realistic image-based referential game.
> > >
> > > My preference would actually be to remove this section entirely: I do not think the general claim is substantiated sufficiently, I do not think the heavily qualified claim (of the generalization phenomena in this dataset) is particularly useful, and I think there are plenty of other interesting experiments in the paper that can fill the remaining space.
> > >
> > > > Q5. More discussion about broader impacts
> > >
> > > This is a great response and the additional experiment is interesting.
> > >
> > > ---
> > >
> > > I believe overall the author response has addressed many of my concerns, especially with the inclusion of the more standard and comparable image-based referential game. However, one outstanding issue is that the generalizability experiments in section 5.3 have *not* been validated by the more standard/difficult experiments, and still remain underexplained. I will very likely raise my score to 6 if
> > >
> > > 1. authors provide precise details of what is tested in the generalizability experiments in sec 5.3 in a response, not delegating the details to supplementary code; and
> > > 2. the general claim of "in complex environments, the emergent protocol generalizes better as the group size increases" is either **heavily qualified**, or eliminated entirely. (Though of course, feel free to disagree with me and convince me how broadly applicable this conclusion is).

---

> > > > ### Author Response · Authors · 2021-08-26
> > > > **Systematicity and generalizability**
> > > >
> > > > Dear Reviewer `c94V`,
> > > >
> > > > We thank you not just for the recent response but for making such great suggestions overall. We sincerely believe that the paper is much more refined with extensive experimental results, the way the paper is presented, and in-depth discussion on the broader impact of our study with added references.
> > > >
> > > > >__Q1. Systematicity and linguistic variation in larger groups__
> > > >
> > > > - First, these are indeed great points to be raised and discussed! We are also very curious whether or to what extent neural agents exhibit properties that are analogous to human linguistic evolution. Further, we believe that there *might* be some clues given from the systematicity paper you suggested ([C]) that *could* explain our findings regarding the generalizability study with neural agents (which we decided to omit entirely, as you suggested).
> > > >
> > > > - Thank you for your clarification. We also believe that findings that confute previous studies are not a *bad* thing. That being said, we think that there needs a clear distinction between *structural (grammatical) systematicity* and *input variability*. With this distinction, we believe that our finding (partially) aligns with what the human language studies have found. From our result with neural agents, we find that: "for larger groups, different agents start to use less similar languages to describe the same observation". This is in direct correspondence to the "linguistic variability" in human languages [O] (or in [C], this corresponds to their finding: *"Importantly, this analysis also confirmed that larger groups were indeed associated with greater input variability overall"*). In fact, human linguistic evolution studies [C, Q, R, S, T] suggest this "input variability" as one of the factors that **promote** "structural systemization" (Quoting Raviv et al. [C]: *"Our results further showed that the increase in structure was driven by the greater input variability in the larger groups."*). Please notice that they indeed explicitly differentiate "variability score" and "structure score" in their analyses (interestingly, Atkinson et al. [P] finds the "opposite" trend, claiming that there is no sign of greater input variability leading to simpler languages).
> > > >
> > > > - Our finding with neural agents that show a "maintained communication success rate" despite increased input variability *might* suggest the emergence of structural systematicity as the group size increases. However, we hesitate to make a definitive statement on this without additional experimental analyses, e.g., the actual measurement of grammatical systematicity (Raviv et al. also measure the communication success rate as *one of* the metrics for such systematicity).
> > > >
> > > > - To summarize, linguistic variability is suggested to be the key factor that promotes structural systematicity in human language studies. Our paper finds higher linguistic variability in larger, artificial neural systems and a *weak* sign of structural systematicity from the maintained communication success rates.
> > > >
> > > > - __Plan for the revision__: In the introduction and the broader impact section, we will add an extensive discussion of how our findings with neural agents are analogous to or confute the human linguistic evolution literature.
> > > >
> > > > - Again, we sincerely appreciate your comments on building the connection between our findings with neural agents and the findings from human languages. We would be more than happy if you have additional thoughts on this.
> > > >
> > > > >__Q2. Imprecise evaluation__.
> > > >
> > > > - First, we would once again like to thank you for suggesting us to make an improvement on this part. We agree that our description of how the messages are categorized (by ourselves: the authors) was informal and done without sufficient verification. Regardless of whether our categorization is right or not, the paper can greatly benefit from introducing a formal description of message categorization followed by quantitative verification on such categorization.
> > > >
> > > > - *"Could authors confirm that they have such analyses and they confirm the groupings of the authors?"*: Yes, we have the experimental results based on the setting that we described in the author response (random sample 100,000 messages and color them based on our proposed categorization to show that messages in the same (proposed) category are clustered together). __This confirms that our proposed categorization is indeed sensible__. We will add this result in the revised version.
> > > >
> > > > >__Q4. Findings on generalizability__.
> > > >
> > > > - We certainly understand and appreciate your concern regarding our findings on generalizability. __We have decided to remove our discussion on generalizability entirely in our paper__. We list our reasoning behind this decision as follows:
> > > >
> > > > 1. __Premature experimental results to make a definitive claim__: We agree that the current experimental setup and results are premature to make definitive claims about generalizability on emergent communication involving artificial agents. Indeed, we found confounding results on generalizability from two different game setups. We acknowledge that there is a big logical leap from these two results to our claim: "In complex (simple) environment emergent communication generalizes better (worse) as group size increases". We still very much believe that there is "something interesting" that can be investigated and analyzed in-depth, but the current development is not enough to be formally presented in the paper. As you pointed out, testing the generalizability in more complex games (real-world images) or adding finer granularities in terms of "simple" and "complex" spectrum would be necessary to substantiate the current claim.
> > > >
> > > > 2. __Existing literature on human languages that can be linked__: Reading papers from the human linguistic studies you suggested, we found interesting studies on the relationship between group size and generalizability. Specifically, there seems to be an active discussion and analysis on group size, input variability, generalization, and systematicity in human languages, and we believe that our findings on increased generalizability in larger groups (in "complex" environments) can be linked with these studies. In [C], authors claim that *"Our main prediction was that larger groups would create more structured languages, given that they are under a stronger pressure for __generalization__ due to increased input variability and reduced shared history"*. Again, we believe that the study on generalizability in relation to group sizes deserves more analyses and extensive literature review.
> > > >
> > > > 3. __Repositioning of the paper__: While addressing the comments from all reviewers, we have repositioned our paper to a great extent. First, we have put more emphasis on the communication graph optimization section with additional experiments using "soft constraints". Second, we have added more game setups using the standard referential games and with the CIFAR-10 dataset. Third, we have more discussions regarding the game setup, comparison with the human linguistic evolution studies, and human social networks. We agree with you that we have sufficient contents to replace the generalization study subsection.
> > > >
> > > > - __Plan for the revision__: We will remove the generalizability subsection entirely and instead discuss it as one prominent future direction in the discussion section.
> > > >
> > > > >__Details on the generalizability experiment__.
> > > >
> > > > - Below, we describe details of the generalizability experiment.
> > > >   1. *Color variations*: each colors have default RBG decimal codes: 'red': (255, 0, 0), 'green': (0, 255, 0), 'blue' : (0, 0, 255), 'cyan' : (0, 255, 255), 'magenta': (255, 0, 255), 'yellow': (255, 255, 0).
> > > >     1-1. Train set: the colors are sampled from RGB codes with random variations of $\pm$ 32.
> > > >     1-2. OOD test set: the colors are sampled from RGB codes with random variations of $\pm$ 64.
> > > >   2. *Radius variations for ellipses*: The longer radius is sampled uniformly in the range of 0.8 -1.2.
> > > >     2-1. Train set: The shorter radius is sampled uniformly in the range of (0.8 - 1.0) * the longer radius.
> > > >     2-2. OOD test set: The shorter radius is sampled uniformly in the range of (0.6 - 1.0) * the longer radius.
> > > >   3. *Angle variations for polygons*: Polygons are chosen from: triangles (n=3), quadrilaterals (n=4), pentagon (n=5), hexagon (n=6).
> > > >     3-1. Train set: Each angle is sampled uniformly in the range of (1.6 $ \pi / n$ - 2.4 $ \pi / n$ ).
> > > >     3-2. OOD test set: Each angle is sampled uniformly in the range of (1.2 $ \pi / n$ - 2.8 $ \pi / n$ ).
> > > >   4. "Simple" vs. "complex" game setting:
> > > >     4-1. "Simple" setting: Colors for objects are chosen from ("red", "green", "blue") and the shapes are chosen from ("triangle", "quadliteral", "ellipse"), each observation contains two objects (therefore, there are two possible decision-actions only).
> > > >     4-2. "Complex" setting: Colors for objects are chosen from ("red", "green", "blue", "cyan", "magenta", "yellow") and the shapes are chosen from ("triangle", "quadliteral", "ellipse", "pentagon", "hexagon"), each observation contains ten objects (therefore, there are ten possible decision-actions).
> > > >
> > > > [O] A.S. Ghyselen, G. De Vogelaer. "Seeking systematicity in variation: Theoretical and methodological considerations on the “Variety” concept". Frontiers in psychology 9, 2018
> > > >
> > > > [P] M. Atkinson, S. Kirby, K. Smith. "Speaker Input Variability Does Not Explain Why Larger Populations Have Simpler Languages". PLoS ONE, 2015
> > > >
> > > > [Q] R.L. Gómez. "Variability and detection of invariant structure". Psychol. Sci. 2002
> > > >
> > > > [R] S. Lev-Ari "How the size of our social network influences our semantic skills". Cogn. Sci. 2016
> > > >
> > > > [S] G.C. Rost, B. McMurray. "Speaker variability augments phonological processing in early word learning". Dev. Sci. 2009
> > > >
> > > > [T] L.K. Perry, L.K. Samuelson, L.M. Malloy, R.M. Schiffer. "Learn locally, think globally: exemplar variability supports higher-order generalization and word learning". Psychol. Sci. 2010

---

> > > > > ### Comment · Reviewer_c94V · 2021-08-26
> > > > > **Thanks**
> > > > >
> > > > > Thanks to authors for this detailed response! I appreciate the detailed discussion on the relation to the human linguistic evolution literature, and the additional details about the generalization experiments. Through this discussion I believe the authors have addressed all of my concerns, and I am raising my score to 7.
> > > > >
> > > > > To be perfectly clear, my vote to remove the generalization results is **not** because I do not believe they are worth pursuing: I think this is a very interesting idea and would encourage the authors to look more into this phenomenon. I simply believe that, as presented, the ideas were not fully fleshed out as they could be, and I am glad to reach a consensus with the authors on this point.

---

> > > > > > ### Author Response · Authors · 2021-08-27
> > > > > > **Thank you!**
> > > > > >
> > > > > > Dear Reviewer `c94V`,
> > > > > >
> > > > > > Thank you for taking a substantial amount of time and effort to review our paper and engage in discussions to give constructive feedback and meaningful comments.
> > > > > >
> > > > > > On the generalization experiment, we agree that it is definitely worth investigating more on this. We are very thrilled to encounter some relevant and interesting studies in the human linguistic evolution literature. Furthermore, we will run more experiments using more diverse sets of image data with different complexity settings. We do agree with you that at the current stage, the study on this part needs to be more developed, especially since we have observed confounding results in two different settings. We are very motivated to dive more into this in the future and make it more convincing and impactful with additional experiments and idea development.
> > > > > >
> > > > > > With the current submission, we have significantly repositioned our paper, putting more emphasis on 1) the communication graph optimization with soft constraints, 2) discussing more on the different game setups with partial and full observations and with real-world CIFAR-10 image datasets, 3) adding more discussions in relation to the human linguistic evolution literature, and 4) formal presentation of the message categorization.
> > > > > >
> > > > > > Again, thank you for the insightful comments. Please let us know if there is anything else that we should address.

---

> > > > > > > ### Author Response · Authors · 2021-08-30
> > > > > > > **Generalizability experiment using CIFAR-10.1**
> > > > > > >
> > > > > > > Dear Reviewer `c94V`,
> > > > > > >
> > > > > > > We found a real-world image dataset, CIFAR 10.1 by Recht et al. [*], that allows testing the OOD generalization ability.
> > > > > > > In the general response [2/2], we ran an additional experiment for the generalizability using this dataset. With this dataset, the generalization performance grew as the group size increased.
> > > > > > >
> > > > > > > With this result along with the finding from the human linguistic evolution literature, we are very interested to investigate more on this. We are still reluctant to add this result to our revised paper (with a lot of repositionings) because we still cannot make a definitive claim on which setting leads to better generalization in larger groups.
> > > > > > >
> > > > > > > Again, we thank you for providing such constructive and insightful comments.
> > > > > > >
> > > > > > > [*] B. Recht, R. Roelofs, L. Schmidt, V. Shankar. Do ImageNet Classifiers Generalize to ImageNet? In ICML, 2019.

---

### Official Review · Reviewer_e6Vg · 2021-07-16

**Rating:** 6
**Confidence:** 4

**Summary:**

The paper investigates the results of Emergent Communication learning in different settings (number of agents, communication connectivity, task difficulty). The authors use a reference game variant where the communicating parties are limited in their perceptual abilities and hence need to communicate in order to solve the task (select a specific object among a number of alternatives). The authors obtain a number of interesting observations (for example, observing that larger groups of agents result in more diverse communication protocols, which (in certain settings) still allow to achieve high communication success rate).

## Strengths
- The authors tackle a highly relevant problem
- The experimental setup aims to investigate the impact of a number of factors that may indeed be crucial in determining the properties of emergent communication
- The paper has a large number of illustrations and is generally well-written, which promotes understanding.

## Weaknesses
- The decision to use early stopping, while motivated by valid concerns, does not actually address these concerns. A universally controlled amount of training in terms of total number of interaction per agent would be much easier to interpret. Ideally, more than one schedule should be tested: 1-training until convergence, 2-training for a set number of interactions for any given agent.
- Literature review overlooks a number of relevant works and thus inadvertently presents the experimental setup as more novel than it is.

**Limitations And Societal Impact:**

The limitations were partially addressed, but I believe that the paper would benefit from expanding that section, especially when it comes to the potential confounding effects of early stopping.

There is no discussion of societal impacts, but I think it is reasonable since the paper is not addressing a question of immediate practical application.

**Main Review:**

## Originality, novelty & relevance

The paper addresses a relevant problem by providing a new take on a well-studied referential game setup.
Although the setup explored in the paper is quite close to a number of existing works, I believe that the paper expands previously known results and modifies the setup enough to be considered sufficiently novel.

## Clarity

The paper is generally well written and is a pleasure to read. The authors also provide a number of high-quality visualizations that further promote understanding and improve the reading experience.

## Literature review

The literature review and, generally, a number of branding decisions severely limit (in my opinion) the quality of the paper.

Firstly, some quotations are not very precise. For example, authors say that they adapted a symmetric setting since it "allows all agents to function as both speakers and listeners and suppresses potential unfavored bias that might arise in the asymmetrical setting that potentially distracts our analyses on the convergence of a shared language", quoting Bouchacourt and Baroni, 2019 (https://arxiv.org/pdf/1905.11871.pdf). But the quoted material does not seem to support the claim. Here is a quote from the cited paper: "However, we also find that even perfectly symmetric agents converge to distinct idiolects instead of developing a single, shared code."

Secondly, certain phrasing decisions sound like the authors claim to introduce things that are, in fact, already thoroughly explored. For example, the authors say "we construct a unique setting within which communication among agents is indispensable for achieving their shared goals", which makes a confusing impression, since such a description is true for most (if not all) referential game settings. Similarly, on line 59, the authors phrase the symmetric nature of their game as a novelty, while it was investigated in numerous existing works.

Perhaps, the most concerning thing is that the literature review, while giving a broad overview of the field, does not discuss in enough detail the works most closely resembling the paper.

For example, the effect of population size on emergent communication outcomes was considered in
https://direct.mit.edu/isal/proceedings/isal2020/32/678/98420

In a similar vein, the authors say "Our work is distinctive in a way that we do not assign agents into hard clusters of communities, and therefore, we observe finer granularities of linguistic continuums at agent-levels rather than at community-levels", but https://arxiv.org/abs/2007.09820 studies the effects of connectivity structure on emergent communication without hard clustering.

Furthermore, the authors "introduce message agreement and communication success rates used to evaluate language convergence", without mentioning https://arxiv.org/abs/1903.05168 or https://direct.mit.edu/isal/proceedings/isal2020/32/678/98420 which both pay particular attention towards discussing metrics that can/should be used to study emergent communication in similar setups.

Finally, the paper repeatedly mentions that it is not injecting architectural bias. E.g. "without injecting architectural bias into the environment or agents that could favor or hamper communication, our goal is to investigate the effect of emergent communication and analyze how agents’ general agreement on the communication protocol changes under varying group settings." This claim is vague, and it seems to slightly misrepresent the situation. For example, one of the limitations of this work is that the agents are trained in a centralized manner (i.e. they "merge" during training to propagate gradients). While this decision is understandable and common, it is unrealistic for modeling real-world communication system evolution, and is an example of a bias that fosters communication learning.

Overall, the literature review helps to place the work in the global context of the field, but omits a number of works that closely resemble the present contribution. Additionally, a number of phrasing decisions are sub-optimal, giving the paper an impression of making unwarranted claims.

## Experiments

Most of the experiments are reasonable and provide insight into the problem in question. There are, however, limitations.

One of the questionable decisions is using the Early Stopping criterion. While the authors mention the problem in the appendix C3, their solution does not seem satisfactory. For example, under their schedule, when N=2, the agents stop training after 1000 epochs. At the same time, for N=128, 60000 epochs is required. On the first glance it seems that each agent in the group adds ~500 epochs to training and that each agent gets the same amount of exposure to training stimuli. But judging by the code, (multiagent_communication.py), the epoch size itself depends on the population size. As a result, agents in larger communities get exposed to a larger set of stimuli, which is crucial for generalization.

This limitation could explain the results in Figure 3. For complex tasks, agents in small communities simply learn too fast to get exposed to a large enough pool of stimuli, hence they don't generalize well.

This seems to contradict what authors wrote in lines 252-260. In particular, they say that "number of observations provided to agents during training is maintained or even decreases depending on the early stopping criteria". It's not clear how this conclusion was reached. I would appreciate a clarification on that matter.

Lastly, some experiments have unclear interpretations. For example, figure 4 claims that dialects only form in the cases with relatively large maximal communication distance. At the same time, it seems that dialects do form in all conditions, but the dialect groups are very small. Additionally, the experiment seems to contradict the claim on line 69-70. The authors state "At the same time, distant pairs of agents that have not interacted during training can still communicate.", while figure 4k strongly suggests that distant agents actually fail to communicate.

## Conclusion

The paper tackles an important problem and provides a number of interesting results.

Many of the key experimental results, however, are either weakened by the early stopping problem or are not as clearly interpreted as authors present them (for example, the qualitative evaluation of what is happening on figure 4).

The literature review does not (in my opinion) properly represent the paper's relation to previous work and does not properly separate novel contributions from previously known results.

At present, I select "marginally below acceptance", but I will be open to read other reviewers' opinions and the authors' rebuttal.

I would be especially glad to be wrong when it comes to the early stopping problem (I described my concern in the "Experiments" section of the review).

## Questions
I would appreciate a comment on my concerns in the Experiments and Literature review sections.

Additionally, for how many epochs were the agents trained to obtain results in Figure 4? Was the "early stopping" criterion applied as well?

## On authors' response

I have read the responses and increased my score by 1 point. The reason I can not select "full accept" is because although authors plan to rewrite parts of the paper and tone down the claims, I can not check the quality of the final product.

**Time Spent Reviewing:**

7

---

> ### Author Response · Authors · 2021-08-12
> **Initial response to Reviewer e6Vg [1/3]**
>
> We thank reviewer `e6Vg` for taking a substantial amount of time to review our manuscript. We sincerely appreciate that you felt pleasure reading our paper. Thank you for the constructive comments. We provide responses to all your comments/questions below:
>
> ---
>
> >__Q1. The early stopping criterion__: *"The decision to use early stopping, while motivated by valid concerns, does not actually address these concerns. [...] the epoch size itself depends on the population size. As a result, agents in larger communities get exposed to a larger set of stimuli, which is crucial for generalization."*:
>
> 1. This is a critical misunderstanding. First, we would like to clarify the early stopping (ES) criterion itself. We used ES in our experiments to prevent us from making claims that are not true but rather byproducts of overfitting/underfitting. Following the general practice, we *early stop* training by looking at __the number of epochs without any progress made__, where the progress is defined by the validation loss. Therefore, the *patience* parameter in the source code (multiagent.py) is __not__ the actual number of training epochs but is the number of epochs after which the training should stop if the progress has not been made. Therefore, the __actual__ training epochs that you pointed out from Figure 7 in the appendix are actually decided by the ES criterion (based on the validation loss), not by the hard threshold.
>
> 2. *"As a result, agents in larger communities get exposed to a larger set of stimuli, which is crucial for generalization."* This is also a misunderstanding. In fact, our current ES criterion, which is set to *n_agent* multiplied by the patience parameter __favors smaller groups__ in terms of reaching the training convergence because while the ES criterion increases *linearly* with respect to *n_agent*, the number of interactions (or, the number of communication links) between agents in a group increases *quadratically* with respect to *n_agent*. As you have pointed out from Figure 7 in the appendix, we observe that regardless of the group size, each agent gets around 500 to 1,000 epochs of training (we, the authors, did not set this number; it automatically derived from the early stopping criterion, or, the validation loss). However, if two agents, one from the group of size 2 and the other from the group of size 128, get the same number of epochs for training, it is very likely that the training of the agent in $N=128$ group is not fully converged, because that agent is receiving much diverse communication signals (or, __linguistic variability__ [F]) from the larger number of speaker agents, while the total number of signals agents in different group get is roughly the same (500 - 1000 each; again, automatically decided from the ES criterion). This is analogous to a situation where two people are exposed to the same amount of learning stimuli, but one person needs to learn just one language while the other needs to learn multiple languages (or, dialects).
>
> 3. *"This limitation could explain the results in Figure 3. [...]"* Because of the points that we made above, the current ES criterion actually __favors__ small group, and the generalization performance of the larger groups is actually __suboptimal__ (while that of the smaller group is very close to the optimum value that we can get from training). To confirm this, we conducted __additional experiments__ with varying *patience* parameters. While we could not find increased generalization performance for $N=2$, we observed __increased__ generalization performance for $N=16$ and larger groups when we increased the patience parameter. Therefore, our claim on the generalizability of group communication is __not__ due to potential overfitting or the training not being converged.
>
> 4. *"Ideally, more than one schedule should be tested: 1-training until convergence, 2-training for a set number of interactions for any given agent."* We respectfully disagree with this comment. The first suggestion, training until convergence, is actually what we aim to do by adopting the early stopping criterion. If we set a hard threshold on the number of interactions/observations per agent, we will inevitably observe either 1) overfitting in smaller groups or 2) underfitting in larger groups, and claims made henceforth will be affected by this.
>
> 5. *"Additionally, for how many epochs were the agents trained to obtain results in Figure 4? Was the "early stopping" criterion applied as well?"* Yes, we used the ES criterion for all our experiments.
>
> - We would be more than happy to extend the discussion if needed.
>
> - __Plan for revision__: In the revised manuscript, we will 1) add more explanation on the early stopping criterion and 2) add additional experimental results.
>
> ---
>
> >__Q2. Incorrect citations__: *"[...] a number of phrasing decisions are sub-optimal, giving the paper an impression of making unwarranted claims."*:
>
> - We acknowledge that our claim on the *symmetry* of the game design removing *architectural biases* is incorrect, and we will remove all our claims regarding this. As you have pointed out, The work of Bouchacourt and Baroni 2019 [A] designs symmetric games to reflect human-like property but this does not warrant the removal of architectural biases. We do find the game design of Graesser et al., 2019 [B] to be similar to ours, as they also endow agents with imperfect evidence that needs to be aggregated via (group) communication to solve the shared task.
>
> - __Plan for revision__: We will make sure that we remove all unwarranted claims and citations. We update the justification of our game setting and explain where this setting stands with respect to the past literature.
>
> ---
>
> >__Q3. Missing citations__: *"[...] certain phrasing decisions sound like the authors claim to introduce things that are, in fact, already thoroughly explored"*:
>
> - We agree and will add these citations. Thanks for suggesting the work of Dubova and Moskvichev, 2020 [C] and Dubova et al., 2020 [D]. We will add discussions regarding the game settings used in their work (RL-based, coordination) in relevance to ours (partial observability and referential), as well as the group scale (up to 6, 10 agents) compared to our setting (up to 256 agents). Finally, we will add discussions on the metrics, referencing [C] and [E].
>
> ---
>
> [A] D. Bouchacourt, M. Baroni. Miss Tools and Mr Fruit: Emergent Communication in Agents Learning about Object Affordances. In ACL 2019
>
> [B] L. Graesser et al., Emergent Linguistic Phenomena in Multi-Agent Communication Games. In EMNLP 2019
>
> [C] M. Dubova, A. Moskvichev. Effects of Supervision, Population Size, and Self-Play on Multi-Agent Reinforcement Learning to Communicate. In Artificial Life Conference Proceedings 2020
>
> [D] M. Dubova, A. Moskvichev, R. L. Goldstone. Reinforcement Communication Learning in Different Social Network Structures. In Workshop on Language in Reinforcement Learning, ICML 2020
>
> [E] R. Lowe et al., On the Pitfalls of Measuring Emergent Communication, AAMAS 2019
>
> [F] A.S. Ghyselen, G. De Vogelaer. Seeking systematicity in variation: Theoretical and methodological considerations on the “Variety” concept. Frontiers in psychology 9, 2018

---

> > ### Comment · Reviewer_e6Vg · 2021-08-28
> > **Thank you for your response**
> >
> > I appreciate a thoughtful response to my review and I apologize for a late reply.
> >
> > Most of my concerns were partially or completely addressed. I hope that authors include the discussion and justification of the early stopping approach in the paper.
> >
> > Since most of my concerns are resolved, I increase my score.

---

> > > ### Author Response · Authors · 2021-08-30
> > > **Early stopping approach**
> > >
> > > Dear Reviewer `e6Vg`,
> > >
> > > Thank you again for providing such insightful and constructive reviews.
> > > We will add discussions regarding the early stopping criterion and add comparisons to the "fixed epochs per agent" approach.

---

> ### Author Response · Authors · 2021-08-26
> **Additional response to Reviewer e6Vg [2/3]**
>
> >__Unclear interpretations__
>
> - Thank you, and we agree that the overall writing especially in section 5.4 can be refined based on your comments.
>   1. *Dialects*: We will rephrase lines 2-4 in Figure 4 (and the corresponding sentences in the main text) as follows: *When the maximal communication distance ranges from 1 (a) to 16 (d), groupings of communications that are peculiar to their locality, e.g., dialects are formed. However, when the maximal communication distance is small (a), dialect clusters are relatively small, and such clusters from distant localities often exhibit high topographic similarity. In contrast, when the maximal communication distance is large (d), the topographic similarity of dialects is clearly reflected by the localities.*
>   2. *Communication success*: We will replace the sentence in lines 60-70 with the following sentence: *"At the same time, distant pairs of agents that have not interacted during training can still communicate __when a certain threshold for the maximal communication distance is reached__"*. Furthermore, we will also rephrase the relevant sentences accordingly in section 5.4.
>
> ---
>
> >__Additional experiment with new game designs__:
>
> - We acknowledge that our claim on the *symmetry* of the game design removing *architectural biases* is incorrect, and we will remove all our claims regarding this. For example, as you (Reviewer `e6Vg`) pointed out, the work of Bouchacourt and Baroni (2019) [A] designs symmetric games to reflect human-like property but this does not warrant the removal of architectural biases. We will make sure that we remove all unwarranted claims and citations.
>
> - We also acknowledge that we have incorrectly emphasized our novelty in the game setup. In fact, there have been numerous experimental setups that share a great degree of similarity to ours, especially on the partial observation. Our misplaced emphasis on the game setup has raised concerns on whether our findings with this "unique" setting can be applied to the existing, more "traditional" settings, and this concern is echoed by other reviewers `c94V` and `8o6e`.
>
> - Our setting is not unique. In fact, partial observation with disjoint and complementary evidence given to different agents has been the key formulation that encourages the emergence of interactive communication and has been adopted by numerous previous research [G,H,I,J]. Crucially, Graesser et al. (2019) [B] studies the emergence of interactive communication in groups with the partial, disjoint, and complementary setting similar to ours.
>
> - The other setting that promotes emergent interactive communication between agents is the fully observed setting. This setting, referred to as the *referential game*, originates from the work of Lazaridou et al. (2017) [K] and is a variant of the *signaling game* (Lewis, 1969 [L]). Here, both agents are given the same, full observations. Communication between agents is encouraged because 1) only one agent (the speaker agent) has access to the *instruction* and 2) the *orders* or the objects displayed in the observations are different by the agents. Therefore, our setup does  __not__ correspond to the referential game, and we will make this point clear in the revised manuscript.
>
> - To show that our findings and claims are applicable for both settings, __we have conducted additional experiments__ with the full-observation setting (i.e., the referential game) and using real-world images. Specifically, the color and shape variations are maintained, but all agents observe the full evidence from such objects (not color-only nor shape-only). Only the speaker agent is provided with the instruction, and the orders of the objects are shuffled in the observations of the speaker and the listener agents. In this setting, our findings are:
>   1. The messages the speaker agents send to the listener agents changes in this setting. In the partial observation setting, the speaker agents send messages to the listener agents to inform the agent about the *positions* about the target object (Appendix D). On the other hand, in the full observation setting, since only the speaker agent has an access to the instruction and the positions of the objects are shuffled for the listener agent, the messages entail information about the *target object itself* (i.e., whether the target object is a blue circle or a red square).
>   2. The findings and the claims made in with the partial observation setting __can also be made in the full observation setting__. Specifically, 1) in all-to-all communication, the message similarity decreases but the success rate is maintained as the group size increases and 2) the emergence of dialects in the neighbor-to-neighbor communication.
>   3. In addition, we have conducted an __additional experiment on a referential game with image-based dataset__. Specifically, we used the same setting as the added (conventional) referential game setting but the objects are __now image-based__ (__CIFAR-10__ [M]) instead of the shape-color variations. Again, our claims regarding the population size and connectivities (sections 5.3 and 5.4) can also be made with the CIFAR-10 dataset.
>
>
> - __Plan for the revision__: First, we will correct all claims and misplaced citations that highlight the uniqueness of our game setup. Then, in the experimental setup section, we will discuss the two commonly adopted setups for interactive emergent communication, which are: 1) partial observation and 2) full observation with instruction given to one agent only (i.e., the referential game). Then, we will state that our setting falls into the former category. Lastly, we will state in the revised manuscript that the claims/findings from the partial observation setting are echoed in the full observation setting, and __include the experimental outcomes from the full observation setting__. Furthermore, we will state in the revised manuscript that the claims/findings from the shape-color-based objects dataset are echoed with the image-based dataset, and __include the experimental outcomes with this dataset__.
>
> ---
>
> [G] J. Foerster, Y. Assael, N. de Freitas, S. Whiteson. Learning to Communicate with Deep Multi-Agent Reinforcement Learning. In NeurIPS 2016
>
> [H] N. Jaques, A. Lazaridou, E. Hughes, C. Gulcehre, P. A. Ortega, D. Strouse, J. Leibo, N. de Freitas. Social Influence as Intrinsic Motivation for Multi-Agent Deep Reinforcement Learning. In ICML 2018
>
> [I] I. Kajic, E. Aygün, D. Precup. Learning to cooperate: Emergent communication in multi-agent navigation. arXiv preprint arXiv:2004.01097, 2020.
>
> [J] S. Sukhbaatar, A. Szlam, R. Fergus. Learning Multiagent Communication with Backpropagation. In NeurIPS 2016
>
> [K] A. Lazaridou, A. Peysakhovich, M. Baroni. Multi-Agent Cooperation and the Emergence of (Natural) Language. In ICLR 2017
>
> [L] D. Lewis. Convention: A philosophical study. 1969
>
> [M] A. Krizhevsky. Learning Multiple Layers of Features from Tiny Images. 2009.

---

> ### Author Response · Authors · 2021-08-26
> **The discussion phase ends in a week**
>
> Dear Reviewer `e6Vg `,
>
> We truly appreciate your time and effort to review our manuscript and think that we have made substantial improvements in our paper thanks to your comments.
>
> This is a __friendly reminder__ to let us (the authors) know if you have any outstanding questions or comments regarding our responses, as the discussion phase ends roughly in a week. We have responded to all your comments and conducted additional experiments that can significantly strengthen our paper.

---

> ### Author Response · Authors · 2021-08-28
> **Additional response regarding the early stopping criterion [3/3]**
>
> Dear Reviewer `e6Vg`, to be absolutely certain about the early stopping criterion issue, we are listing the actual numbers of training interactions given per agent during the group communication when the early stopping criterion is applied (currently reported results in the paper are derived from these training epochs). In addition, we have conducted additional experiments as you suggested: __giving the same amount of training stimuli for different group sizes__.
>
> >__Q1. Additional response on the actual number of training interactions given per agent__: *"This seems to contradict what authors wrote in lines 252-260. [...] It's not clear how this conclusion was reached. I would appreciate a clarification on that matter."*:
>
> - From our clarification on how our early stopping criterion is actually applied, we hope that you are now clarified on the fact that the ES criterion does *NOT* favors larger groups. Below, we provide the actual number of training interactions given per agent during the group communication when the early stopping criterion is applied. We confirm the actual training stimuli given per agent does NOT increase as the group size increases:
>
> | Group size             	| 1   	| 2   	| 4    	| 8    	| 16  	| 32  	| 64  	| 128 	|
> |------------------------	|-----	|-----	|------	|------	|-----	|-----	|-----	|-----	|
> | # Interactions / agent 	| 684 	| 582 	| 1088 	| 1024 	| 703 	| 609 	| 658 	| 500 	|
>
> >__Q1. Additional experiment: training for a set number of interactions for any given agent__:  *"A universally controlled amount of training in terms of total number of interaction per agent would be much easier to interpret. Ideally, more than one schedule should be tested: 1-training until convergence, 2-training for a set number of interactions for any given agent."*:
>
> - We have conducted additional experiments with a fixed number of training epochs given per agent for all group sizes. From the table above (using the ES criterion), we have fixed the number of training epochs per agent to 1) 500 and 2) 1,000 and conducted all-to-all and generalization experiments. Below are the results:
>
> *Table H. Generalization performance (comm. success rate) when # Interactions / agent = 500.*
>
> | Group size    | 1   	| 2   	| 4    	| 8    	| 16  	| 32  	| 64  	| 128 	|
> |-------------	|-----	|-----	|------	|------	|-----	|-----	|-----	|-----	|
> | Success rate 	| 72.3 	| 72.7 	| 73.2 	| 73.4 	| 73.5 	| 73.5 	| 73.6 	| 73.5 	|
>
> *Table I. Generalization performance (comm. success rate) when # Interactions / agent = 1,000.*
>
> | Group size    | 1   	| 2   	| 4    	| 8    	| 16  	| 32  	| 64  	| 128 	|
> |-------------	|-----	|-----	|------	|------	|-----	|-----	|-----	|-----	|
> | Success rate 	| 72.0 	| 72.5 	| 73.3 	| 73.3 	| 73.6 	| 73.6 	| 73.6 	| 73.7 	|
>
> - From the results, we confirm that larger groups are still better at generalization via communication. In fact, fixing the number of epochs per agent (i.e. NOT applying the ES criterion) does *not* seem to affect the results in a significant manner (and notice the *increased* generalization performance for the larger groups when the fixed epoch is set to 1,000). We additionally confirm that __we were able to make the same claim regarding section 5.3__: *as group size increases, messages become less similar but the communication success rate remains*, using the fixed number of interactions as well.
>
> - At this stage, we sincerely hope that we have addressed your concern regarding the ES criterion. We still believe that adopting the ES criterion is enough to deliver our claims on emergent group communication. However, we are curious to receive your feedback on this. If you think that adopting the ES criterion is not enough to make convincing claims on our findings, we will put the above results and figures that reproduce other group communication experiments that are currently in the paper.

---

### Official Review · Reviewer_z6vN · 2021-07-17

**Rating:** 6
**Confidence:** 4

**Summary:**

The paper studies the effect of population size and graph connectivity on the emergence of communication. The main questions studied in the paper were 1) how does the size of agents' population affect task success? and 2) when agents don't get to train directly, but through some other agents, how well they can communicate with each other when they put in contact? The game that was used in this paper to study the problem, was a referential game, and the task was discrimination task. The agents can take 2 roles of speaker and listener, such that speaker gets some observation and send a message to listener, and listener gets some other observation along with speaker's message, and it has to make a decision in a discriminative setup.

**Limitations And Societal Impact:**

I have 3 main concerns:
1. Centralized training: Although we see in the litrature that centralized training may be an acceptable approach in some cases, I believe for the problem understudied in this paper, we need to train agents independently. The authors argued that only gradients flow from listener agent to the speaker agent, and it doesn't contain information about parameter values of the other agent. I believe gradients contain information about inner state of the listener agents such that without having this information, training speaker agents may be hard! The nature of emergent communication research require independent training as the message sent by speaker shouldn't be tuned based on inner state of listener, but only by the task reward.
2. Supervised training: did you try to train agents using reinforcement learning as well?
3. Some of the conclusions are not clear to me. For example, the authors showed communication success graphs in Figure 2 and 3. Is it the results after you trained the agents and took the best performing ones? I don't know how to interpret the results shown as I believe the entire training should be shown (at each timestep). In Figure 2, the fact that population size doesn't affect communication success rate may mean all the agents in the population are copies of each other! Is that the case?

**Main Review:**

The paper is well-written and well-organised.

The problem understudy is an interesting and can be an impactful problem to study. The idea of the paper is not novel and it was studied in previous papers. Also, the game that was designed is a referential game that you can find very similar games in other emergent communication papers.

I found the methodology used to solve the problem, i.e. train agents, not a very good approach! There are some weaknesses when it comes to the used methodology. One of them is centralized training! In multi-agent systems, using centralized training violates a fundamental assumption that information should not flow between agents. The other weakness is related to using a discrimination loss and treat the problem as a supervised learning rather than reinforcement learning. The discrimination loss is a very strong signal such that training such agents will be an easy task.


**Time Spent Reviewing:**

6

---

> ### Author Response · Authors · 2021-08-16
> **Initial response to Reviewer z6vN**
>
> Thank you for taking the time and effort to review our paper. We respond to all your questions and questions below:
>
> ---
>
> >__Q1.Centralized training.__ *"In multi-agent systems, using centralized training violates a fundamental assumption that information should not flow between agents.", "The nature of emergent communication research require independent training as the message sent by speaker shouldn't be tuned based on inner state of listener, but only by the task reward."*:
>
> - We respectfully disagree with your comment.
>
> - First, __your comment on centralized training is not specifically targeted to our unique formulation and contribution, but rather can be applied to all emergent communication studies that adopt the centralized training & decentralized execution (C&D) paradigm__.
>   - *"In multi-agent systems, using centralized training violates a fundamental assumption that information should not flow between agents."*: Centralized training does __not__ "violate the fundamental assumption" in "multi-agent systems". The paper by Foerester et al. (2016) [A], a seminal work on emergent communication, claims that "Centralised planning and decentralized execution" is "a standard paradigm for multi-agent planning.", citing [B] and [C]. In their work, centralized training is suggested as an __alternative__ to the decentralized training methods, as they yield significantly better empirical results.
>   -  There has already been a __vast amount of previous research__ [H, I, J, K, L, M, N, O], and even in group communication setting [P] that adopts the C&D paradigm in emergent communication literature, further countering your claim that *"The nature of emergent communication research requires independent training [...]"*.
>   - Reviewer `e6Vg` also believes that the C&D paradigm is "understandable and common".
>   - According to the review paper of the emergent communication [D], the idea of decentralized training in emergent communication is closely linked with the desire to acquire messages that are discrete, as it "provides the symbolic scaffolding for interfacing the agents’ emergent code to natural language, which is universally discrete" [E].
>     - In our work, we DO study emergent group communication with discrete symbols, under the C&D paradigm, using the Gumbel-softmax relaxation [F, G]. Again, we are not the first to use the C&D paradigm + discrete messages in the emergent communication literature, as there is also another study on emergent communication with the C&D paradigm + discrete messages [H, I, J].
>     - Other than this motivation, we do believe that there can be some potential issues with the centralized training not being applicable to certain tasks or environments because of the inter-gradient flow during agents. However, we believe that our study's applicability is NOT blocked by the centralized training idea (although we are not aware of any actual cases within which the C&D paradigm is not applicable). It is actually quite the opposite, as for example, our study on optimizing the communication graph (section 5.5) is NOT possible with the decentralized training method because of the large training time and the instability that accompanies the RL methods with a large number of agents.
>   - Our work constructs a non-embodied multi-agent system that comprises up to __hundreds__ of agents. To the best of our knowledge, this is the first work that scaled the number of agents up to this size, both in centralized and decentralized learning (planning). On the contrary, the previous emergent group communication formulates agent groups of sizes less than 10 or, 30 at most [P, Q, R, S]. Given the vast amount of emergent communication studies under the C&D paradigm (listed above), we believe that scaling it up to large groups of hundreds of agents can itself be a unique contribution.
>   - __Plan for the revision__: We will add an extensive discussion on the centralized and decentralized training as well as the C&D paradigm and further highlight that our work follows the C&D paradigm.
>
> ---
>
> >__Q2. Additional experiments using reinforcement learing__. *"did you try to train agents using reinforcement learning as well?"*:
> - __We have conducted additional experiments with reinforcement setting__. For group sizes up to 16, we trained agents with discrete messages using the REINFORCE algorithm in a decentralized setting, with a regularization term to encourage diverse messages (following [Q, T]). The claims we made using the centralized, supervised training are echoed with the RL setting.
> - __Plan for the revision__: We will add the experimental results using the decentralized, RL training.
>
> ---
>
> >__Q3. Experimental results.__ *"Is it the results after you trained the agents and took the best-performing ones?", "[...] may mean all the agents in the population are copies of each other"*:
> - This is a __serious misunderstanding__. First of all, we did __not__ choose the best-performing agents, nor the best results. All results shown in the paper are based on a __test set__ after the agents are trained until convergence. All agents in a population are __NOT__ copies of each other. They do not share weights whatsoever and are endowed with their own trainable weights. The messages transmitted from different agents all differ (e.g., Figure 2 (b)) directly counters your concern.
>
>  We would be more than happy to be further engaged in this discussion. Please let us know if there's any misunderstanding of your comments on our end.
>
> ---
>
> [A] J. Foerster, Y. Assael, N. de Freitas, S. Whiteson. Learning to Communicate with Deep Multi-Agent Reinforcement Learning. In NeurIPS 2016.
>
> [B] L. Kraemer, B. Banerjee. Multi-agent reinforcement learning as a rehearsal for decentralized planning. Neurocomputing, 2016.
>
> [C] F. A. Oliehoek, M. T. J. Spaan, N. Vlassis. Optimal and approximate Q-value functions for decentralized POMDPs. JAIR, 2008.
>
> [D] A. Lazaridou, M. Baroni. Emergent multi-agent communication in the deep learning era. arXiv preprint arXiv:2006.02419, 2020.
>
> [E] C. Hockett. The origin of speech. Scientific American, 1960.
>
> [F] E. Jang, S. Gu, B. Poole. Categorical reparameterization with Gumbel-softmax. In ICLR, 2017.
>
> [G] C. Maddison, A. Mnih, Y. W. Teh. The concrete distribution: A continuous relaxation of discrete random variables. In ICLR, 2017.
>
> [H] S. Havrylov, I. Titov. Emergence of language with multi-agent games: Learning to communicate with sequences of symbols. In NeurIPS, 2017.
>
> [I] R. Lowe, Y. Wu, A. Tamar, J. Harb, P. Abbeel, I. Mordatch. Multi-agent actor-critic for mixed cooperative-competitive environments. In NeurIPS, 2017.
>
> [J] S. Sukhbaatar, A. Szlam, R. Fergus. Learning multiagent communication with backpropagation. In NeurIPS, 2016.
>
> [K] D. Kim, S. Moon, D. Hostallero, W. Kang, T. Lee, K. Son, Y. Yi. Learning to schedule communication in multi-agent reinforcement learning. In ICLR, 2019.
>
> [L] A. Singh, T. Jain, S. Sukhbaatar. Learning when to communicate at scale in multiagent cooperative and competitive tasks. In ICLR, 2019.
>
> [M] A. Das, T. Gervet, J. Romoff, D. Batra, D. Parikh, M. Rabbat, J. Pineau. 2019. Tarmac: targeted multi-agent communication. In ICML, 2019.
>
> [N] J. Jiang, Z. Lu. Learning attentional communication for multi-agent cooperation. In NeurIPS, 2018.
>
> [O] S., David, N. Lau, L. P. Reis. Multi-agent actor centralized-critic with communication. Neurocomputing, 2020.
>
> [P] O. Tieleman et al., Shaping representations through communication: community size effect in artificial learning systems. In NeurIPS ViGIL 2019
>
> [Q] L. Graesser et al., Emergent Linguistic Phenomena in Multi-Agent Communication Games. In EMNLP 2019.
>
> [R] M. Dubova, A. Moskvichev. Effects of Supervision, Population Size, and Self-Play on Multi-Agent Reinforcement Learning to Communicate. In Artificial Life Conference Proceedings 2020.
>
> [S] M. Dubova, A. Moskvichev, R. L. Goldstone. Reinforcement Communication Learning in Different Social Network Structures. In Workshop on Language in Reinforcement Learning, ICML 2020.
>
> [T] K. Evtimova, A. Drozdov, D. Kiela, and K. Cho. Emergent communication in a multi-modal, multi-step referential game. In ICLR, 2018.

---

> > ### Comment · Reviewer_z6vN · 2021-09-01
> > **Thank you for your response**
> >
> > Thank you for your response regarding my concerns. Centralized training in a multi-agent system is a limitation for modelling real-world problem, the fact that it was used in previous literature doesn't make it the best training regime. I understand for some environments/tasks that training independent RL agents is hard or impossible, decentralized training be used as a heuristic/hack. If this is the case in this paper, I appreciate an explanation on why decentralized training is the best approach here.
> >
> > Thanks for your response on the test set. It will be beneficial if you include the performance of agents during training as well, i.e. how quickly and to what extends agents learn during training (collective task success). It is informative to see how long you trained the agents in the population, and how much is the variance in agents' performance in the population (during training) in different network architectures.
> >
> > You responded to the rest of comments. Thank you.

---

> > > ### Author Response · Authors · 2021-09-02
> > > **Discussion on the centralized training and the performance during training**
> > >
> > > Dear reviewer `z6vN`,
> > >
> > > Thank you for the feedback. We will add discussions on different training regimes as well as the additional results using RL for training.
> > > We will also include our analyses on the predictive performance during training.

---

> ### Author Response · Authors · 2021-08-26
> **The discussion phase ends in a week**
>
> Dear Reviewer `z6vN`,
>
> Thank you for reviewing our paper and giving constructive comments/feedback.
>
> This is a __friendly reminder__ to let us (the authors) know if you have any outstanding questions or comments regarding our responses, as the discussion phase ends roughly in a week. We have responded to your comments and conducted additional experiments that can significantly strengthen our paper.

---

### Official Review · Reviewer_8o6e · 2021-07-19

**Rating:** 7
**Confidence:** 4

**Summary:**

This paper studies population size and connectivity graph effects in emergent communication in a population of agents. By limiting the communication partners in specific ways, i.e. deleting edges from the fully connected graph, the authors show that dialects can evolve: communication protocols that are specific to sub-groups even when those groups are not completely isolated. Furthermore, by experimenting with different connectivity patterns that share a constant overall graph sparsity, they find that a pattern with few highly connected nodes (hubs) leads to significantly higher communication success than random or uniform patterns. Such hubs are a common feature of complex networks found in many natural systems. In terms of communicative success, the authors find that more complex tasks are associated with better generalization to unseen data (a smaller drop in success relative to the training data)

The paper also introduces a variant of the widely used referential game where the speaker and listener observe non-overlapping aspects of the images they communicate about, and where agents act as both listener and speaker, depending on how they are sampled at the start of a round in the game.


**Limitations And Societal Impact:**

Yes, section 6 and appendix G.

**Main Review:**

The signalling game variant proposed in the paper is interesting, in that it has different information requirements than the more common fully observed one. The speaker does not know the target exactly, and has to communicate to the listener which images are candidates. One aspect that did not become obvious to me while reading the paper is whether the listener also receives the instruction, or whether the speaker has to communicate that too.

To better understand the contribution of the partial observation, it would be good to ablate it by replicating some of the experiments in a fully observed signalling game, and investigating whether the resulting language patterns are at all different. A priori I don’t see a very clear reason why they should be, but I would be happy to be proven wrong on this point.

The graph configuration effects are nice results. The emergence of dialects even in the absence of clear subgroups (with higher internal connectivity than external) like in figures 4b and 4c is even somewhat surprising. The optimal graph question is a valuable contribution the field of multi-agent communication, as it has the potential to enable a significant saving in the computational cost of such experiments.

One question that is left open, but would be very interesting and probably relatively straightforward to answer, is what happens to the optimal graph structure when the constraint that each node has at least three outgoing vertices is dropped. The result found here, with the hub nodes, follows one pattern that often occurs in natural complex systems, but goes against another - that of the power law distribution. I would have expected to see the power law here too; can you comment on why it does not emerge?

The results you find are mostly about communicative success, but within the population that the agents are trained with. I would really like to know whether the languages that are formed in such non-uniform connectivity graphs have the same properties as those from uniform graphs when it comes to compositionality and learnability. E.g. analysing the topographic similarity of the messages to the ground truth feature vectors could provide some insights here, as could training newly initialized evaluation agents against members of a fully trained population.

A third way to make the paper stronger would be the inclusion of more standard datasets, to enable easier comparisons to existing work.

All in all, I believe this paper takes on a very interesting and relevant question, but is a little light on the results presented.

**Time Spent Reviewing:**

4

---

> ### Author Response · Authors · 2021-08-13
> **Initial response to Reviewer 8o6e**
>
> We thank reviewer `8o6e` for taking the time and effort to review our paper and providing insightful comments. We are also very glad that you appreciated and are positively surprised by 1) our results on the communication graph optimization (or, the emergence of *hub* nodes) and 2) dialects in the absence of subgroups. Below, we answer all your individual comments and questions.
>
> ---
>
> >__Q1. Optimal graph structure__: *"[...] would be very interesting [...] is what happens to the optimal graph structure when the constraint that each node has at least three outgoing vertices is dropped", "[...] I would have expected to see the power law here too"*:
>
>  - We appreciate your comment on our experimental setup and results. We also believe that our result on the communication graph optimization has the potential to contribute to saving communication costs in multi-agent communication in various tasks. At the same time, there are limitations on our experimental setup, especially 1) the __hard threshold on the number of outgoing communication links__ and 2) __excessive computational cost and training time__ for optimization. In fact, the experimental results presented in the paper with just 32 (16 + 16 bipartite) agents is an outcome of a week of training time with 128 compute cores. The small group size with the hard constraint on the number of communication links resulted in a relatively __dense communication graph__, which we believe is the reason why the power-law distribution did not emerge. To further investigate the problem, we have extended our experiment based on your suggestion:
>  	- __Additional experiment__ without the limit on the number of outgoing communication links: First, when we completely remove the constraint on the number of outgoing links per agent, the optimized communication graph became extremely dense (i.e., almost became the all-to-all graph). Then, in an attempt to negotiate the trade-off between hard-threshold and no-threshold-at-all, we endowed *soft-constraints* in the number of communication links based on agents' localities by penalizing the formation of communication links between distant agents. From this modified experimental setup, we have observed clearer power-law distribution along with the hub nodes, in much resemblance to real-world human graphs (social networks). __We will add this additional result in our revised manuscript__.
>
> - __Plan for the revision__: We will add a new experimental setup and results. When the communication graph is optimized with __soft constraints__, i.e., penalizing the formation of communication links by the distance between agents, the communication graph that more resembles the real-world human social networks with power-law distribution has emerged. In addition, the rest of the findings in the previous results, i.e., the emergence of hub nodes and much-improved communication success rate (which is the optimization target itself) can still be found in the additional experiment.
>
> ---
>
> >__Q2.Game setup with partial observation__: *" [...] would be good to ablate it by replicating some of the experiments in a fully observed signaling game, [...]", "[...] way to make the paper stronger would be the inclusion of more standard datasets [...]"*
>
> - First of all, we acknowledge that we have incorrectly emphasized our novelty in the game setup. In fact, there have been numerous experimental setups that share a great degree of similarity to ours, especially on the partial observation. Our misplaced emphasis on the game setup has raised concerns on whether our findings with this "unique" setting can be applied to the existing, more "traditional" settings, and this concern is echoed by another reviewer `c94V`.
>
> - Our setting is not unique. In fact, partial observation with disjoint and complementary evidence given to different agents has been the key formulation that encourages the emergence of interactive communication and has been adopted by numerous previous research, [B,C,D,E] to just name a few. Crucially, Graesser et al. (2019) [A] studies the emergence of interactive communication in groups with the partial, disjoint, and complementary setting similar to ours.
>
> - The other setting that promotes emergent interactive communication between agents is the fully observed setting. This setting, referred to as the *referential game*, originates from the work of Lazaridou et al. (2017) [F] and is a variant of the *signaling game* (Lewis, 1969 [G]). Here, both agents are given the same, full observations. Communication between agents is encouraged because 1) only one agent (the speaker agent) has access to the *instruction* and 2) the *orders* or the objects displayed in the observations are different by the agents. Therefore, our setup does  __not__ correspond to the referential game, and we will make this point clear in the revised manuscript.
>
> - To show that our findings and claims are applicable for both settings, __we have conducted additional experiments__ with the full-observation setting (i.e., the referential game). Specifically, the color and shape variations are maintained, but all agents observe the full evidence from such objects (not color-only nor shape-only). Only the speaker agent is provided with the instruction, and the orders of the objects are shuffled in the observations of the speaker and the listener agents. In this setting, our findings are:
>   1. The messages the speaker agents send to the listener agents changes in this setting. In the partial observation setting, the speaker agents send messages to the listener agents to inform the agent about the *positions* about the target object (Appendix D). On the other hand, in the full observation setting, since only the speaker agent has an access to the instruction and the positions of the objects are shuffled for the listener agent, the messages entail information about the *target object itself* (i.e., whether the target object is a blue circle or a red square).
>   2. The findings and the claims made in with the partial observation setting __can also be made in the full observation setting__. Specifically, 1) in all-to-all communication, the message similarity decreases but the success rate is maintained as the group size increases and 2) the emergence of dialects in the neighbor-to-neighbor communication.
>   - We did not conduct the experiment on the communication graph optimization with the full observation setting.
>
> - __Plan for the revision__: First, we will correct all claims and misplaced citations that highlight the uniqueness of our game setup. Then, in the experimental setup section, we will discuss the two commonly adopted setups for interactive emergent communication, which are: 1) partial observation and 2) full observation with instruction given to one agent only (i.e., the referential game). Then, we will state that our setting falls into the former category. Lastly, we will state in the revised manuscript that the claims/findings from the partial observation setting are echoed in the full observation setting, and __include the experimental outcomes from the full observation setting__.
>
> ---
>
> >__Q3. Additional analyses__. *"[...] the languages that are formed in such non-uniform connectivity graphs [...]", "analyzing the topographic similarity of the messages to the ground truth feature vectors"*:
>
> - Thanks for the suggestions! We do believe that these can be interesting to analyze. We would be thrilled to pursue this direction in future research, along with others suggested in our discussion section.
>
> ---
>
> [A] L. Graesser, K. Cho, D. Kiela. Emergent Linguistic Phenomena in Multi-Agent Communication Games. In EMNLP 2019
>
> [B] S. Sukhbaatar, A. Szlam, R. Fergus. Learning Multiagent Communication with Backpropagation. In NeurIPS 2016
>
> [C] J. Foerster, Y. Assael, N. de Freitas, S. Whiteson. Learning to Communicate with Deep Multi-Agent Reinforcement Learning. In NeurIPS 2016
>
> [D] N. Jaques, A. Lazaridou, E. Hughes, C. Gulcehre, P. A. Ortega, D. Strouse, J. Leibo, N. de Freitas. Social Influence as Intrinsic Motivation for Multi-Agent Deep Reinforcement Learning. In ICML 2018
>
> [E] I. Kajic, E. Aygün, D. Precup. Learning to cooperate: Emergent communication in multi-agent navigation. arXiv preprint arXiv:2004.01097, 2020.
>
> [F] A. Lazaridou, A. Peysakhovich, M. Baroni. Multi-Agent Cooperation and the Emergence of (Natural) Language. In ICLR 2017
>
> [G] D. Lewis. Convention: A philosophical study. 1969

---

> > ### Comment · Reviewer_8o6e · 2021-08-18
> > **Thanks for the response**
> >
> > Thank you for your extensive response!
> >
> > The soft constraints experiment sounds like a great addition - the human networks that produce power law connectivities also tend to have such constraints, due to the cost of maintaining connections between people.
> >
> > Thanks for doing the fully observed experiments and clarifying the role of the fully or partially observed nature of the game, as well as the CIFAR-10 experiment. Could you provide the results here for the discussion?
> >
> > With the additional experiments and the repositioning of the partially observed game, I feel the paper is both more balanced and more complete. I’m raising my score to a 7.
> >
> > For the benefit of future readers I would recommend doing the graph optimization experiment in the full observation setting if possible, since full observation is more standard in the literature, and the graph effects seem to be independent of the observation, as you found. Combining the partial observability and the graph optimization into one result requires readers to hypothesize about how to disentangle the two.

---

> > > ### Author Response · Authors · 2021-08-21
> > > **New experiments and results**
> > >
> > > Dear Reviewer `8o6e`,
> > >
> > > Thank you for carefully reading our response. We sincerely appreciate your comments on the optimal graph structures and game setups. We also believe that after the new experiments we conducted as well as the repositioning of the game setup, the paper will be much more clear and complete. Please find the additional experimental results summarized in the general response at the top.
> > >
> > > Thank you again for suggesting graph optimization with the full observation setting. We are currently running the experiment for this and will discuss/update the results when it's finished.

---

> > > > ### Author Response · Authors · 2021-08-30
> > > > **Communication graph optimization with the full observation setting**
> > > >
> > > > Dear Reviewer `8o6e`,
> > > >
> > > > We would like to let you know that we have actually conducted another experiment that you suggested, which was to run the communication graph optimization experiment with the full observation (more common setting of the referential game). We generated figures and tables that correspond to Table 1 and Figure 5 in the paper using this setting and showed the optimized success rates in Table J of the general response (2/2). We will add this result to the revised manuscript.
> > > >
> > > > Again, thank you for your invaluable comments that have greatly contributed to increasing the quality of our submission!

---

### Author Response · Authors · 2021-08-20
**General response [1/2]**

## General response

We thank all the reviewers for the constructive reviews. Below, we present a part of the result from the additional experiments that can be presented in the table format. The additional experimental results including these tables and more with figures will be added to the revised manuscript.

---

### Communication graph optimization with soft constraints

As a follow-up to section 5.5 of our paper: communication graph optimization, we did a new experiment. In the previous experiment with the hard threshold, the number of outgoing links per agent was fixed. Now, the objective function penalizes the formation of communication links between distant agents: communication success rate subtracted by the penalty term $k * (| i - j | + 1)$ with $i$ and $j$ being agents' localities. Here, $k$ is a parameter that regulates the importance of the penalty term in the optimization. We set $k = 0.0012$ so that the objective value becomes zero in the all-to-all communication (fully connected graph).  With this setting, the new optimized communication graph is drastically different from the previous one. Most agents are now connected with nearby agents, with a small number of links from distant pairs. The formation of few distant links suggests that these links are crucial for optimization (i.e., increasing the overall communication success rate) despite the large penalty involved.

*Table A. Number of agents with different in-degrees in the optimized communication graph with soft constraints.*

| In-degree 	| 1  	| 2 	| 3 	| 4 	| 5 	| 6 	|
|-----------	|----	|---	|---	|---	|---	|---	|
| # Agents  	| 18 	| 7  	| 3  	| 2  	| 1  	| 1  	|

From this additional experiment, we will show:

1. The emergence of a scale-free graph. In this optimized graph, the in-degree of agents follows the power-law distribution  (Table A).

2. Hub-nodes. We will present an optimized communication graph in the same format as Figure 5 (d) in the paper.

3. Additionally, we will show 4 snapshots of the communication graph during the optimization process: figures that depict the evolution of a communication graph, from random to scale-free.

---

### Results on the new game setups / RL

In our paper, we designed a game setting where agents only get partial complementary observations so that communication is essential to achieve the shared task. We have conducted additional experiments using the referential game setting, where both speaker and listener gents get access to the full information but only the speaker agent receives the instruction. Here the orders of the objects displayed are different for the speaker and the listener agents. From this setting, we use 1) the shape-color varied objects dataset and 2) image-based datasets (CIFAR-10).

In addition, we conducted experiments with reinforcement learning. For group sizes up to 16, we trained agents with discrete messages using the REINFORCE algorithm in a decentralized setting.

The new experimental results are:

*Table B. Full observation, shape-color objects dataset: message similarity.*

| Group size 	| 2     	| 4     	| 8     	| 16    	| 32    	| 64    	| 128   	|
|------------	|-------	|-------	|-------	|-------	|-------	|-------	|-------	|
| Similarity 	| 0.848 	| 0.820 	| 0.807 	| 0.802 	| 0.791 	| 0.764 	| 0.759 	|


*Table C. Full observation, shape-color objects dataset: Communication success rate.*

| Group size 	| 2     	| 4     	| 8     	| 16    	| 32    	| 64    	| 128   	|
|------------	|-------	|-------	|-------	|-------	|-------	|-------	|-------	|
| Success rate 	| 0.985 	| 0.992 	| 0.991 	| 0.985 	| 0.987 	| 0.989 	| 0.992 	|



*Table D. CIFAR-10: message similarity.*

| Group size 	| 2     	| 4     	| 8     	| 16    	| 32    	| 64    	| 128   	|
|------------	|-------	|-------	|-------	|-------	|-------	|-------	|-------	|
| Similarity 	| 0.904 	| 0.896 	| 0.885 	| 0.888 	| 0.882 	| 0.873 	| 0.859 	|


*Table E. CIFAR-10: Communication success rate.*

| Group size 	| 2     	| 4     	| 8     	| 16    	| 32    	| 64    	| 128   	|
|------------	|-------	|-------	|-------	|-------	|-------	|-------	|-------	|
| Success rate 	| 0.892 	| 0.891 	| 0.895 	| 0.897 	| 0.889 	| 0.895 	| 0.894 	|


*Table F. Decentralized training: message hamming distance.*

| Group size 	| 2     	| 4     	| 8     	| 16    	|
|------------	|-------	|-------	|-------	|-------	|
| 1 - Hamming dist. 	| 0.955 	| 0.942 	| 0.901 	| 0.884 	|


*Table G. Decentralized training: Communication success rate.*

| Group size 	| 2     	| 4     	| 8     	| 16    	|
|------------	|-------	|-------	|-------	|-------	|
| Success rate 	| 0.969 	| 0.963 	| 0.964 	| 0.966 	|


1. The findings made in the partial observation setting (section 5.3) can also be made in the full observation setting / using decentralized training. Specifically, in all-to-all communication, the message similarity decreases but the success rate is maintained as the group size increases (Tables B, C, D, E, F, G).

2. Emergence of dialects: as in section 5.4 (Figure 4 (d) ), we will present the figure of dialects that emerged in the full observation game setting.

3. Different message categorizations. The messages the speaker agents send to the listener agents changes in this setting. In the partial observation setting, the speaker agents send messages to the listener agents to inform the agent about the positions of the target object (Appendix D). On the other hand, in the full observation setting, since only the speaker agent has an access to the instruction and the positions of the objects are shuffled for the listener agent, the messages entail information about the target object itself (i.e., whether the target object is a blue circle or a red square).

---

> ### Author Response · Authors · 2021-08-30
> **General response [2/2]**
>
> Below we update the results for the additional experiments reviewers have suggested.
> We would like to thank the reviewers again for their insightful and constructive comments.
>
> ---
>
> ### Graph optimization experiment in the full observation setting (Reviewer `8o6e`)
>
> We generated figures and tables that correspond to Table 1 and Figure 5 in the paper using the full observation setting (i.e. the referential game) instead of the partial setting. Below we show the communication success rates of the optimized graph and other comparison settings.
>
> *Table J. Communication graph optimization in the full observation setting.*
>
> |              	| All-to-one 	| N-to-N 	| Rand. Links 	| Optimal 	|
> |--------------	|------------	|--------	|-------------	|---------	|
> | Success Rate 	| 0.436      	| 0.487  	| 0.591       	| __0.902__   	|
>
> ---
>
> Generalization experiment with real-world images (Reviewer `c94V`)
>
> We found a real-world image dataset, CIFAR 10.1 by Recht et al. [*], that allows testing the OOD generalization ability.
> With this dataset, the generalization performance still increased as for the larger groups (Table K). Please compare this result with *Table E: CIFAR-10* and *Figure 3: Generalization using shape-color objects data with partial setting* in the paper.
>
> *Table K. Generalization performance using CIFAR 10.1 dataset for testing the generalizability*
>
> | Group size   	| 1    	| 2    	| 4    	| 8    	| 16   	| 32   	| 64   	| 128  	|
> |--------------	|------	|------	|------	|------	|------	|------	|------	|------	|
> | Success rate 	| 74.6 	| 75.3 	| 75.6 	| 75.8 	| 76.0 	| 76.0 	| 76.1 	| 76.2 	|
>
> [*] B. Recht, R. Roelofs, L. Schmidt, V. Shankar. Do ImageNet Classifiers Generalize to ImageNet? In ICML, 2019.

---

### Decision · Program_Chairs · 2021-09-27

**Decision:**

Accept (Poster)

**Comment:**

This paper studies the effect of population size and connectivity of the communication topology of those populations in emergent communication and referential games. While population size has received some (very limited) attention in the literature, how the topology of the communicating agents is organized has been quite understudied. The authors report both communicative accuracy as a function of populations/connectivity but also present analysis of the resulting languages.

Overall, this has been among the most productive authors-reviewers interaction I've experienced. Authors have worked with reviewers to clarify questions and address concerns. Some new content has been generated during the discussion period and so I would ask the authors to update their manuscript respectively to reflect these discussions.